# Sensorimotor computation underlying phototaxis in zebrafish

Sébastien Wolf[1,2], Alexis M. Dubreuil[3], Tommaso Bertoni[1,2], Urs Lucas Böhm [4,5,6,7], Volker Bormuth[1,2], Raphaël Candelier[1,2], Sophia Karpenko[1,2], David G.C. Hildebrand [8,9,11], Isaac H. Bianco[10], Rémi Monasson[3] & Georges Debrégeas[1,2]

Animals continuously gather sensory cues to move towards favourable environments. Efficient goal-directed navigation requires sensory perception and motor commands to be intertwined in a feedback loop, yet the neural substrate underlying this sensorimotor task in the vertebrate brain remains elusive. Here, we combine virtual-reality behavioural assays, volumetric calcium imaging, optogenetic stimulation and circuit modelling to reveal the neural mechanisms through which a zebrafish performs phototaxis, i.e. actively orients towards a light source. Key to this process is a self-oscillating hindbrain population (HBO) that acts as a pacemaker for ocular saccades and controls the orientation of successive swim-bouts. It further integrates visual stimuli in a state-dependent manner, i.e. its response to visual inputs varies with the motor context, a mechanism that manifests itself in the phase-locked entrainment of the HBO by periodic stimuli. A rate model is developed that reproduces our observations and demonstrates how this sensorimotor processing eventually biases the animal trajectory towards bright regions.

---

[1] Sorbonne Universités, UPMC Univ. Paris 06, UMR 8237, Laboratoire Jean Perrin, F-75005 Paris, France. [2] CNRS UMR 8237, Laboratoire Jean Perrin, F-75005 Paris, France. [3] Laboratory of Theoretical Physics, Ecole Normale Supérieure, CNRS, PSL Research University, Sorbonne Universités UPMC, 24 rue Lhomond, 75005 Paris, France. [4] Institut du Cerveau et de la Moelle Epinière, 75013 Paris, France. [5] UPMC Univ. Paris 06, 75005 Paris, France. [6] Inserm UMR 1127, 75013 Paris, France. [7] CNRS UMR 7225, 75013 Paris, France. [8] Program in Neuroscience, Department of Neurobiology, Harvard Medical School, Boston, MA 02115, USA. [9] Department of Molecular and Cellular Biology, Harvard University, Cambridge, MA 02138, USA. [10] Department of Neuroscience, Physiology & Pharmacology, University College London, London WC1E 6BT, UK. [11] Present address: Laboratory of Neural Systems, Rockefeller University, New York, NY 10065, USA. Sébastien Wolf and Alexis M. Dubreuil contributed equally to this work. Correspondence and requests for materials should be addressed to G.D. (email: georges.debregeas@upmc.fr)

To survive and thrive, motile organisms use sensory cues to navigate towards environments where they are more likely to avoid predators, obtain food or find mates. Efficient goal-directed locomotion requires closed-loop coordination between motor action and sensory perception. Each movement induces a new sensory signal, which in turn modulates the forthcoming motor output. This mechanism is at play in a number of goal-directed behaviours, in organisms ranging from bacteria[1] to nematodes[2, 3] and insects[4, 5], but also among humans[6]. Numerous models have been proposed to account for this complex-coordinated motion, but to date no data are available to understand how these behavioural strategies might be implemented at the circuit level in the vertebrate brain.

Here, we take advantage of the accessibility of zebrafish larvae to whole-brain imaging[7–9] to investigate the neural sensorimotor computation underlying phototaxis. This behaviour, which drives the animal towards illuminated regions, is already present at 5 days post-fertilisation and is thus likely to be hard-wired in the larval brain[10–12]. To obtain information regarding the direction of a light source, larvae use two complementary strategies: stereo-visual comparison[12] and spatio-temporal sampling[13]. The first mode relies on the difference in instantaneous perceived intensity at two different angles to infer the direction of the source. In the second mode, the illumination spatial gradient is extracted from two successive samplings obtained before and after a gaze-reorienting movement. This latter approach requires that the visual and motor information are integrated in a timely fashion.

We thus postulated the existence of a central neural circuit that would drive spontaneous gaze shift while integrating unilateral and bilateral changes in illumination so as to bias reorienting bouts towards a light source. We first establish, through behavioural assays, that the gaze and turning-bout orientations are robustly coordinated, and that the statistics of gaze orientation are biased towards illuminated regions. We then use volumetric functional imaging and optogenetic activation to identify a bilaterally distributed neuronal ensemble in the rostral hindbrain that appears to be a strong candidate for the control of spontaneous gaze dynamics. We further investigate how the self-oscillatory dynamics of this population are modulated by

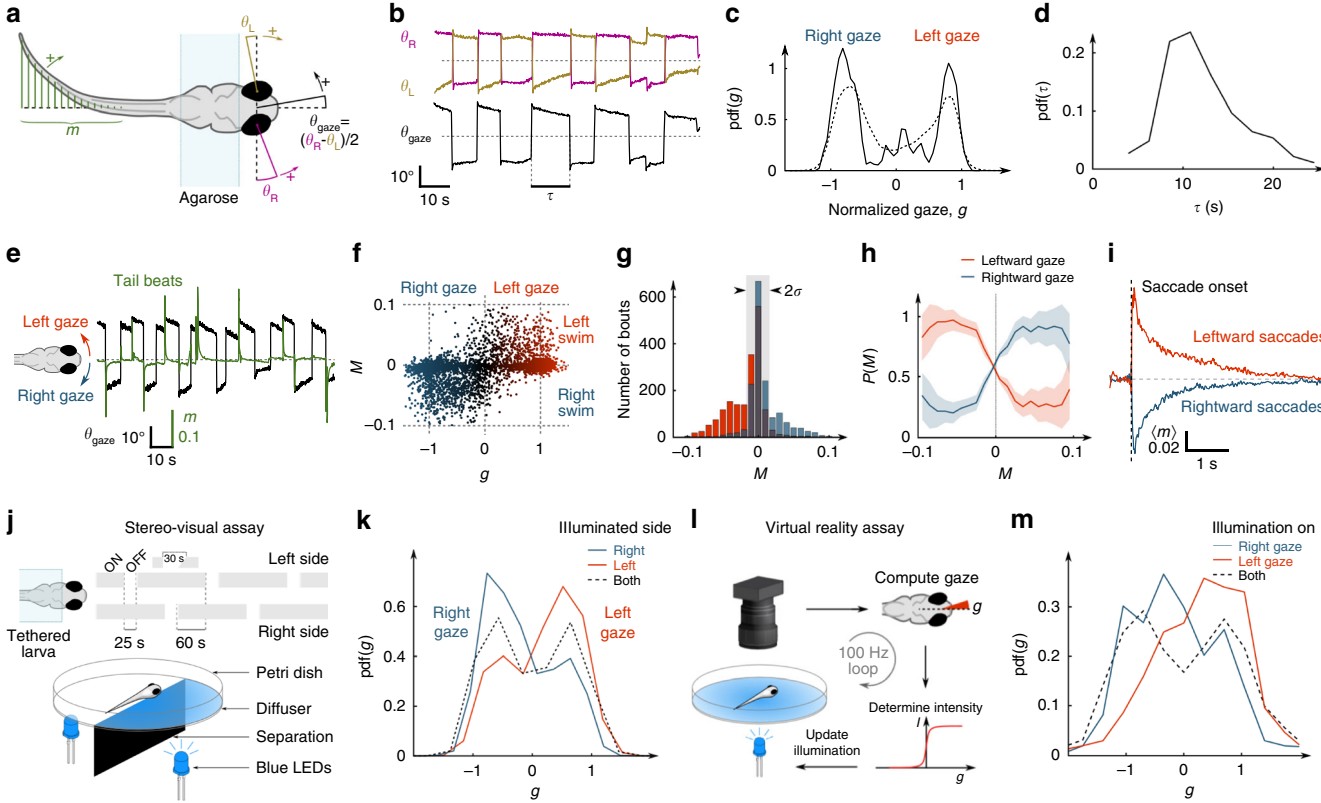

**Fig. 1** Behavioural assays of ocular-saccade–turning-bout coordination and light-induced gaze bias. **a** Definition of the eye and tail kinematic parameters. The larva's tail and eyes are free. The gaze angle, $\theta_{gaze}$, is defined as the mean orientation of both eyes. The parameter $m$ characterises the instantaneous tail deflection (Supplementary Methods). **b–d** Gaze dynamics. **b** Example time-traces of the eye and gaze angles. **c** Probability distribution function (PDF) of the gaze angle, normalised by its characteristic range (Supplementary Methods), for one fish (*solid line*) and for $N = 29$ fish (*dashed line*). **d** PDF of the delay $\tau$ between successive reorienting saccades for one fish. **e–i** Ocular-saccade–tail-beat coordination. **e** Gaze angle and tail deflection signals. **f** Individual tail-beats turning score $M$, defined as the integral of $m(t)$ over the swim-bout (Supplementary Methods), vs. the normalised gaze angle $g$ (3681 tail-beats, $N = 11$ fish). **g** Histograms of $M$ for leftward (*red*) and rightward (*blue*) gaze orientation. The central part of the distribution (standard deviation $\sigma$) is used to assess the significance of the tail-bout orientational bias. **h** Conditional probability of the gaze to be orientated to the left (*red*) or right (*blue*) given the tail-beat turning score $M$ (3681 tail-beats, $N = 11$ fish). The *shaded region* corresponds to the s.e.m. **i** Mean peri-saccadic tail deflection signal averaged over leftward (*blue*) and rightward (*red*) saccades. **j, k** Stereo-visual phototaxis. **j** Scheme of the experimental assay. **k** PDF of the normalised gaze during periods of unilateral stimulation for animals displaying positive phototaxis ($N = 18$ fish). *Red* and *blue curves* correspond to illumination on the left and right eye, respectively. *Dashed curve* indicates bilaterally symmetric illumination. **l, m** Spatio-temporal gaze phototaxis. **l** Scheme of the virtual-reality assay. The fish is submitted to a uniform illumination whose intensity is driven in real-time by the animal's gaze angle. **m** PDF of the normalised gaze angle for virtual leftward (*red*) and rightward (*blue*) illumination ($N = 13$ fish). The *dashed curve* corresponds to the neutral runs (constant illumination)

bilateral and unilateral changes in illumination. Finally, we develop a rate model of this circuit that reproduces most of our observations and provides the first comprehensive description of how phototaxis can be implemented in the vertebrate brain.

## Results

**Visually induced modulation of gaze shift dynamics.** Larvae can redirect their gaze by triggering coherent angular excursions of both eyes—a process called a saccade—or through whole-body reorientation. We examined the endogenous temporal structure

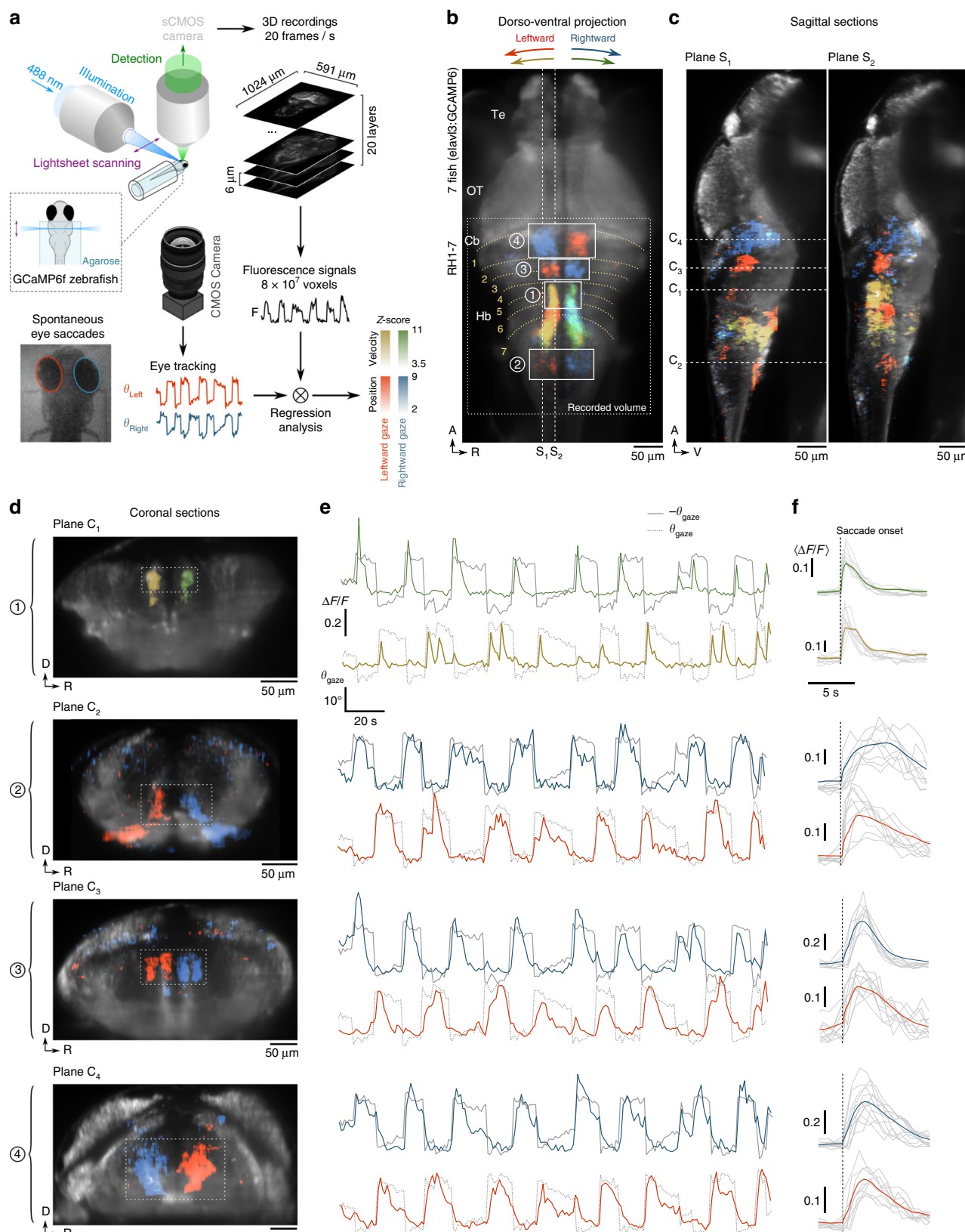

of both processes by monitoring tail deflections and eye orientations in the absence of visual cues in larvae partially restrained in agarose. Spontaneous ocular saccades in zebrafish larvae are highly stereotyped (Fig. 1a–d; Supplementary Fig. 1), with a quasi-periodic alternation between leftward and rightward gaze periods. This property is reflected by the quasi-bimodal distribution of gaze orientations and a distribution of delays between successive reorientations peaked at ~12 s ($12.2 \pm 3.5$ s, mean $\pm$ std, $N = 29$). The orientations of the gaze and the turning bouts were found to be strongly correlated (Fig. 1e–i; Supplementary Fig. 1 and Supplementary Movie 1). We observed that over 85% of the tail-beats that could be unequivocally classified as an attempted turning bout were oriented in the direction of the gaze. Moreover, a large fraction (60.2%) of these bouts occurred within the first second following the onset of an ipsiversive saccade. This coordination mechanism was true for both large-amplitude reorienting saccades that drove the right–left alternation of gaze direction, and for low-amplitude secondary saccades that maintained the gaze to the right or to the left by compensating for slow ocular drift (Supplementarty Fig. 1d–g).

These observations suggest that a unique command circuit may drive ocular saccades and set the direction of successive tail-beats, such that the gaze sequence may be viewed as a proxy for fish reorienting dynamics. We thus posited that phototaxis should manifest as a bias in the animal gaze-angle distribution towards illuminated regions. We tested this hypothesis by replicating, in agarose-restrained animals, two visual stimulation protocols known to evoke phototaxis in freely swimming larvae while monitoring gaze dynamics[12, 13]. Visual stimulation was obtained by projecting two LEDs onto a screen below the animal, each delivering uniform illumination to one eye's field of view (Supplementary Methods).

We first submitted the larva to a 1 min period of whole-field (bilateral) illumination followed by a step decrement of the intensity to one eye, whereas illumination to the other eye was held constant (Fig. 1j; Supplementary Fig. 2). The unilateral illumination period was maintained for 30 s, and the sequence was then repeated. In 67% of the fish (18 of 27), we observed that the gaze orientation was significantly biased towards the more illuminated side during unilateral illumination periods (Fig. 1k). This process was reflected in the trial-averaged post-extinction gaze sequence, which exhibited a transient drift toward the illuminated region (Supplementary Fig. 2f). This bias was associated with a $27 \pm 3.4\%$ (mean $\pm$ std) increase in fixation time towards the illuminated area. In total, a significant phototactic behaviour was observed in 77% of the fish, yet 10% displayed negative phototaxis, in accord with the experiments performed in freely swimming configurations[12]. We confirmed that, owing to gaze–tail coordination, this light-induced gaze bias resulted in a turning bias of the animal toward the light by

performing similar experiments while simultaneously monitoring both the eyes and tail movements (Supplementary Fig. 2k–m).

In a second approach, we developed a virtual-reality phototaxis assay in which the animal had access to spatial cues through spatio-temporal probing only. Both eyes were exposed to the same whole-field illumination at an intensity that was locked in real-time to the gaze orientation (Fig. 1l; Supplementary Fig. 2i–j). The illumination was thus at its maximum when the gaze was oriented to one side, and at its minimum for the other side. A significant light-induced bias of the gaze orientation was observed in 85% of the fish ($p < 0.05$, $N = 11$ of 13). All responsive larvae showed an increased fixation time towards the brighter direction, consistent with a positive phototactic behaviour (Fig. 1m).

In both assays, the imposed spatial light gradient did not trigger a systematic and immediate gaze shift. Instead, it induced a statistical bias of the spontaneous saccadic dynamics whose net result was to increase the relative duration of gaze fixations towards the brightest region.

**Functional mapping of gaze-tuned neuronal populations**. These observations suggested that the saccadic dynamics play a central role in phototaxis by enabling a sequential sampling of illumination gradients through spontaneous gaze shifts, while driving the reorienting dynamics via gaze–tail coordination. We thus sought to identify the saccade command circuit. Ocular saccades in zebrafish have been studied mostly as a model of neural integration[14–16]. The accessibility of this vertebrate model to optogenetic techniques allows one to examine with unmatched precision how a transient neural command that rapidly brings the eyes into a desired position can be transformed into a sustained signal to maintain a fixed eye position[17]. However, in contrast with mammals[18], the upstream command circuit that drives the rhythmic alternation between leftward and rightward saccades in zebrafish remains elusive[19].

Our approach to identify the saccade command neuronal population consisted of recording large fractions of the brain using light-sheet functional imaging in animals expressing calcium reporter in nearly every neuron ($Tg(elavl3{:}GCaMP6f)$) while monitoring eye dynamics (Fig. 2a; Supplementary Movie 2). We restricted our analysis to the hindbrain and the caudal midbrain, where saccade-induced distortions of the brain tissues remained small enough to enable consistent signal extraction. We implemented a multilinear regression approach to classify individual voxels based on their tuning with respect to the angular position and velocity signals of the eyes[20] (Supplementary Methods). We then applied a non-rigid registration method to align these functional maps onto a single reference brain obtained by averaging four different samples imaged using the

**Fig. 2** Regression-based identification of gaze-tuned neuronal populations. **a** Schematic of the experimental setup and regression analysis. Volumetric recordings on GCaMP6f-expressing larvae were performed using one-photon light-sheet imaging (20 sections per stack, 1 stack per second) while monitoring saccadic dynamics. Voxel-by-voxel regression with the eye orientation signals were used to produce position-tuned and velocity-tuned 3D maps. Notice that the two maps overlap in a small subset of voxels (the two 3D maps are displayed separately in Supplementary Movie 3). **b, c** Dorso-ventral projection view and sagittal sections along two planes of the 3D functional map (mean over 7 fish) showing neuronal populations whose activity is tuned to the gaze orientation (*blue* and *red*) and to the gaze angular velocity (*green* and *yellow*). The voxel colour encodes the Z-score values obtained through multilinear regression (Supplementary Methods). Te, telencephalon; OT, optic tectum; Cb, cerebellum; Hb, hindbrain; RH, rhombomere. The *grey dotted rectangle* indicates the effective recorded volume. **d** Coronal sections along the *dotted lines* shown in (**c**) for one sample of the four regions delineated in (**b**). Region 1 encompasses the saccade generator burst neurons (SGBN); region 2 is the velocity-position neural integrator (VPNI); region 3 and 4 constitute the newly identified gaze-tuned rostral hindbrain population. It consists of four bilaterally symmetric clusters tuned to the ipsiversive gaze orientation (region 3) and 2 more rostral clusters tuned to the contraversive gaze angle (region 4). **e** Example $\Delta F/F$ time-traces for these four regions. The *red* and *yellow* (respectively *blue* and *green*) traces correspond to the sub-populations tuned to leftward (respectively rightward) gaze orientation. The *grey lines* are the gaze angle traces ($-\theta_{gaze}$: solid line; $\theta_{gaze}$: dashed line). **f** Corresponding mean peri-saccade $\Delta F/F$ signals, computed over the leftward (*red*) and rightward (*blue*) saccades. *Grey lines* are peri-saccadic signals for individual saccades

one-photon light-sheet microscope at high spatial resolution. A zebrafish brain atlas[21] was finally used to anatomically localise the various neuronal clusters that displayed robust tuning with the gaze position and velocity (Fig. 2b, c; Supplementary Movie 3).

The most prominent velocity-tuned neuronal assemblies were located in two bilaterally symmetric clusters in rhombomeres 4–6 (rh 4–6, Fig. 2d–f, region 1). These populations correspond to the previously identified saccade generator burst neurons[22] (SGBN). Each lateral cluster triggers ipsiversive saccades through direct activation of two oculomotor nuclei (abducens and oculomotor nucleus III). Position-tuned neurons

were found in the caudal hindbrain in a region consistent with the previously described velocity-position neural integrator[20] (VPNI, Fig. 2d–f, region 2). These analysis allowed us to further discover the existence of position-tuned clusters in the rostral hindbrain (rh 2–3) in the form of four stripes bilaterally distributed on each side of the midline (Fig. 2d–f, region 3). These clusters systematically displayed increased and prolonged activity after every ipsiversive saccade, whether reorienting or secondary (Supplementary Fig. 3). We also found two symmetric clusters, lying at the rostral border of the hindbrain (rh 1) ventral to the cerebellum, that were strongly correlated with contraversive gaze orientation (Fig. 2d–f, region 4). Notice that the

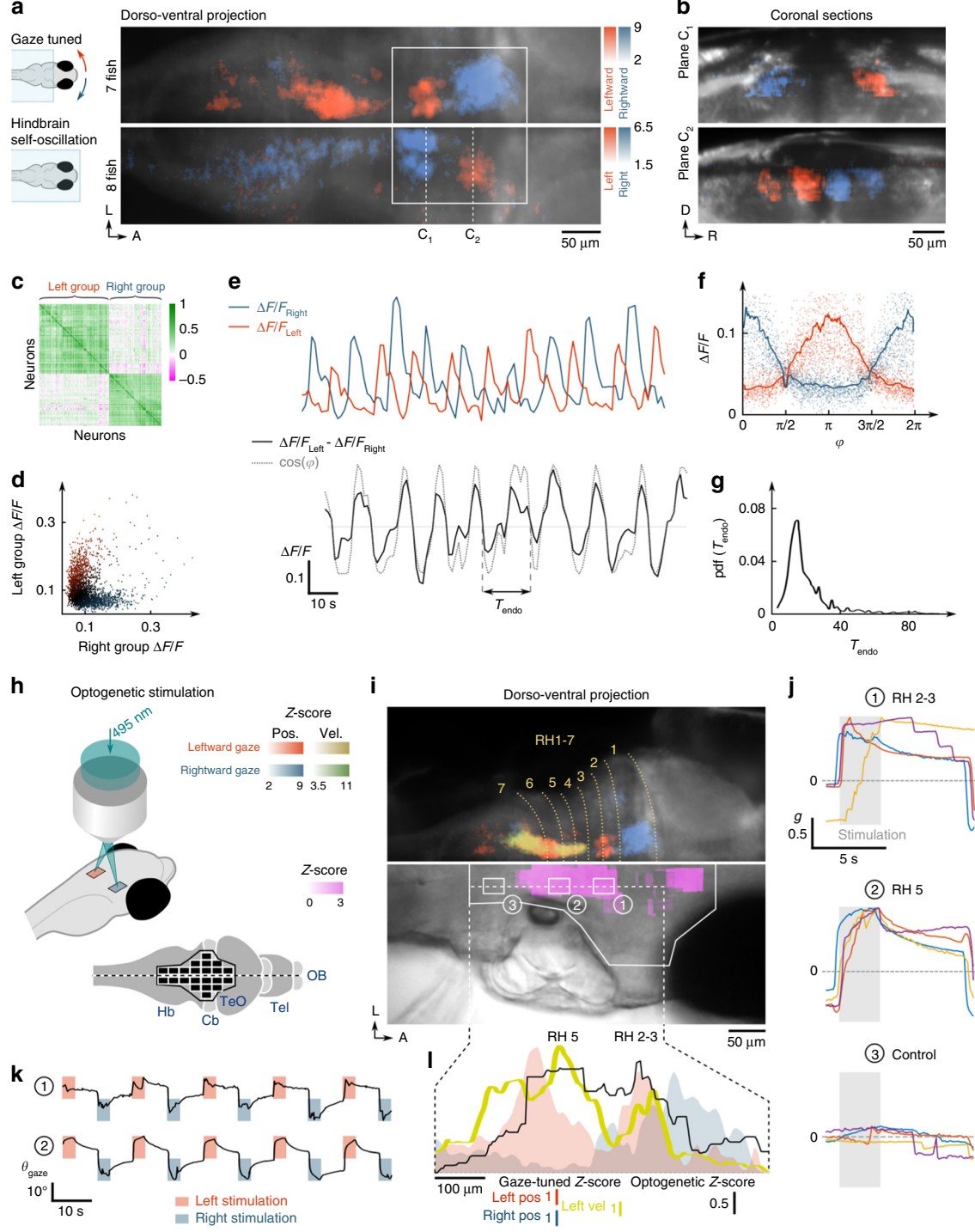

same gaze-tuned neuronal clusters in the rostral hindbrain were independently identified by Alex Ramirez and Emre Aksay in a yet unpublished study.

**Identifying a putative saccade command circuit.** The morphology and anatomical location of this newly identified gaze-tuned ensemble in the rostral hindbrain (the four stripes located in rh 2–3) appeared very similar to the self-oscillating circuit identified by Ahrens et al. in eyes-fixed experiments based on activity correlation analysis[23] (the so-called hindbrain oscillator, or HBO). Consistent with these findings, we observed that this particular neuronal population displayed sustained antiphasic dynamics, with each side activated in alternation, in recordings where the saccades were abolished through eye fixation in agar (Fig. 3a–g). To properly delineate the neuronal population engaged in these self-generated oscillations, which we refer to as the HBO, we first approximated the oscillatory signal using a small subset of neurons and then computed the tuning of each voxel to this reference trace (Supplementary Methods; Supplementary Fig. 4). This functional criterion yields a neuronal map that essentially encompasses the gaze-tuned circuit in rh 1-3 as formerly identified through regression with the gaze signal (Fig. 3a, b; Supplementary Fig. 5). In more caudal regions of the hindbrain (SGBN and VPNI areas, rh4-7), a small number of neurons also participated in these self-oscillatory dynamics. We analysed the temporal structure of the HBO's endogenous oscillations in eye-fixed conditions by running the Hilbert transform on the differential signal (left minus right circuit activity, Fig. 3e–g). We found the period distribution to be peaked at ~20 s ($19.9 \pm 4.8$ s mean ± std, $N = 8$ fish), i.e. close to the spontaneous saccadic period.

The quasi-periodic nature of the saccadic dynamics suggests the existence of a circuit, akin to a central pattern generator, capable of maintaining oscillations with a period on the order of 20 s in the absence of rhythmic sensory or proprioceptive inputs. The HBO constitutes the best candidate for this saccade command circuit. This putative role was found to be consistent with optogenetic assays using larvae expressing ChR2 pan-neurally (Fig. 3h–l; Supplementary Fig. 6). We delivered series of 2.75 s-long light pulses over $47 \times 57 \ \mu m^2$ brain areas, alternately on either side of the midline with a 18 s period (one pulse every 9 s), while monitoring saccades (Fig. 3j). Each pair of bilaterally symmetric regions was tested five times before the targeted regions were relocated to eventually probe a large fraction of the hindbrain encompassing both the HBO and the SGBN. As previously reported[22], activation of the SGBN (rh 4–6) consistently entrained the saccadic dynamics (8 fish of 19, see Supplementary Methods

for details on the quantification of the response). For these responsive fish, activation of the hindbrain in the rh 2–3 region evoked ipsiversive saccades with comparable efficiency (Fig. 3l). These responses could not result from direct activation of the oculomotor nuclei, which lie immediately rostral to rh 1 but drive contraversive saccades. In contrast, optogenetic activation of the two most rostral regions of the oscillator, located in rh 1, failed to evoke conjugated saccades, which might indicate that these two assemblies lie downstream of the HBO's gaze-driving neurons. Finally, a small area in the caudal midbrain was found to trigger ipsiversive saccades in two larvae.

Careful examination of the functional imaging data in eyes-free preparations offers hints regarding a putative connectivity map between the HBO and the SGBN. We identified two small clusters in rh 7 systematically activated 2–3 s prior to the saccade onset (Supplementary Fig. 3a–c), which are thus analogous to the so-called long-lead burst neurons in the mammalian saccadic circuit[24]. Such a functional trait can be understood within the assumption that they receive inhibitory inputs from the contralateral HBO circuit, whose spike-rate consistently decays before the saccade onset. In contrast, the ipsilateral HBO circuit remained silent until the saccade onset, which precludes the possibility that it drives, via excitatory inputs, these long-lead burst neurons. Our observations further indicate that the HBO may receive ascending ipsilateral signals from saccadic pre-motor centres (efference copy). When two successive saccades occurred in the same direction, the second one was systematically followed by a rebound of the HBO active module (Supplementary Fig. 3d, e). This process cannot result from the reciprocal inhibitory coupling with the contralateral HBO module[25], as the latter remains silent during such phases. Although more work will be needed to definitely establish that the HBO drives spontaneous saccades, these various observations suggest a neural architecture of the saccadic command circuit as sketched in Supplementary Fig. 3f.

**The hindbrain oscillator responds to visual stimuli.** Dunn et al.[25] recently reported that the HBO left and right activity biased the orientation of fictive turning bouts to the right and the left, respectively. This finding is in line with our hypothesis that a unique command circuit controls both turning-bout orientation and spontaneous ocular saccades. The resulting gaze–tail coordination process, analogous to the well-documented head–eye coordination in vertebrates[26, 27], explains why the orientation of successive turning bouts exhibits a ~10 s autocorrelation decay in freely swimming larvae[13], as this value corresponds to the typical period between successive reorienting saccades.

**Fig. 3** Self-oscillatory dynamics and optogenetically evoked saccades. **a** Dorso-ventral projection of the gaze-tuned (*top-half*, $N = 7$ fish) and self-oscillatory hindbrain population (*bottom-half*, $N = 8$ eye-fixed fish). In the latter, the *colour* encodes tuning to the left (*red*) and right (*blue*) pre-selected neuronal clusters, revealing strong antiphasic activity (Supplementary Methods). The two functional maps were registered on the same reference brain to enable side-by-side comparison. The *rectangle* indicates the region of the rostral hindbrain gaze-tuned population. **b** Coronal sections of the self-oscillatory population along the *dotted lines* in the projected view. **c** Pearson correlation matrix of the neurons engaged in the self-oscillatory dynamics (616 neurons). The matrix was reordered to reveal two highly correlated (and reciprocally anti-correlated) clusters. **d** Activity of the left vs. right populations ($r = -0.43$). **e** Example $\Delta F/F$ traces of the left and right groups (*top*), and of the differential signal (*bottom*). The *grey dotted line* shows $\cos(\varphi(t))$, where $\varphi(t)$ is the oscillatory phase extracted using Hilbert's transform. **f** $\Delta F/F$ of the right (*blue*) and left (*red*) circuits as a function of the oscillatory phase. The *blue* and *red lines* correspond to the mean values. **g** PDF of the instantaneous oscillation period. **h–k** Optogenetic activation of ocular saccades. **h** Schematic of the optogenetic stimulation protocol. **i** *Top*: projective view of the previously mapped gaze-tuned regions. *Bottom*: Z-score map of saccadic entrainment by optogenetic activation averaged over 8 fish (Supplementary Methods). **j** Mean peri-stimulation normalised gaze signal for three regions: the rostral HBO (rh 2–3), the SGBN (rh 5), a control region (rh 7). The responses of four different fish are shown, and the associated targeted areas are indicated in (**i**). The 2.75 s-long stimulation periods are indicated by the *grey area*. **k** Example gaze signals upon periodic left or right optogenetic stimulation for two region pairs in rh 2–3 and rh 5. **l** Profile plot of the mean optogenetic Z-score along the rostro-caudal axis (*black*), overlaid on the ipsiversive (*red*) and contraversive (*blue*) gaze-tuned Z-score *yellow curve* indicates the ipsiversive velocity-tuned Z-score. Ipsiversive saccades are evoked with comparable efficiency by targeting the SGBN (rh 5) or the rostral HBO (rh 2–3)

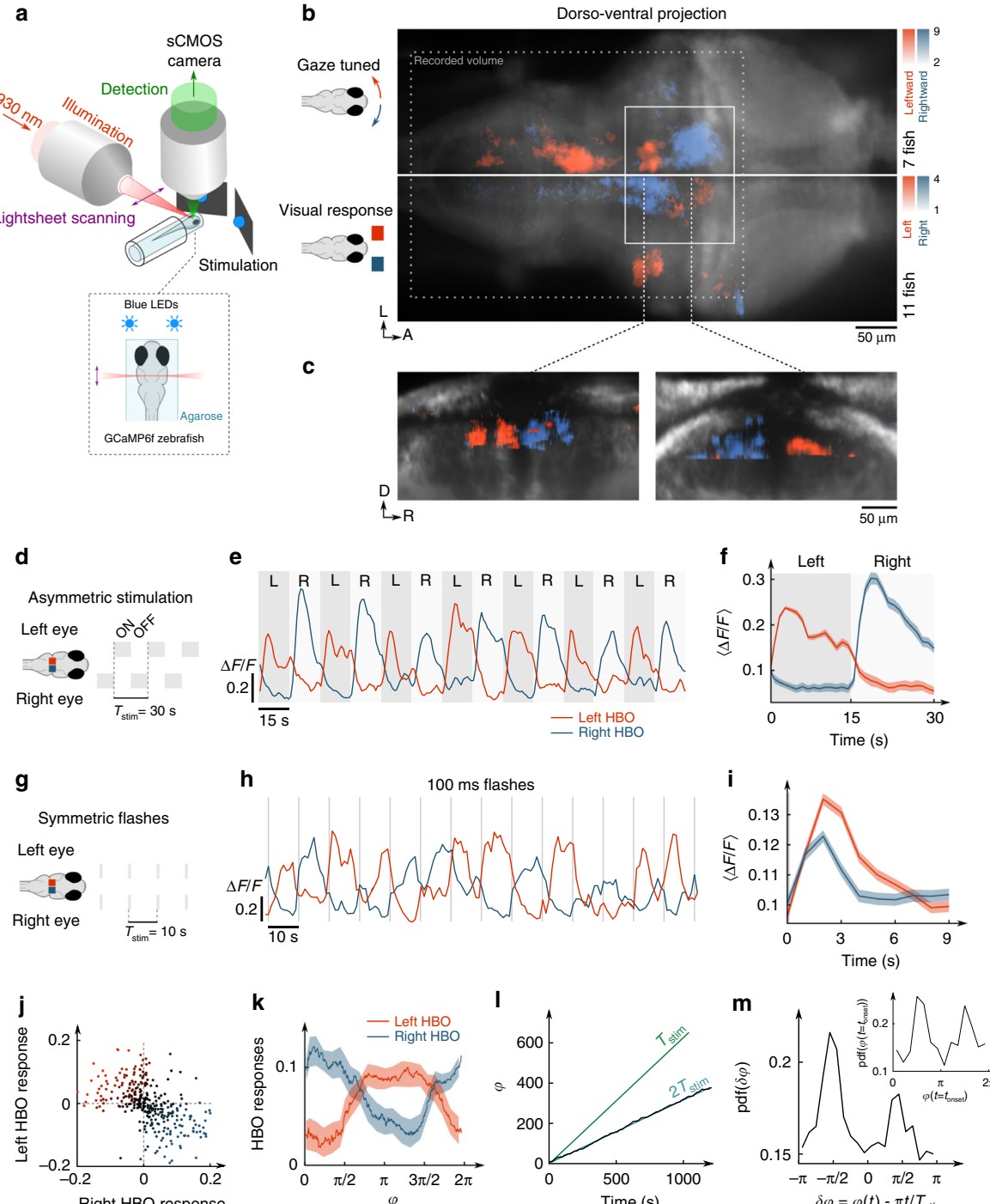

**Fig. 4** Response of the HBO to asymmetric and symmetric visual stimuli. **a** Schematic of the two-photon light-sheet imaging setup with stereo-visual stimulation. **b** Comparison of the gaze-tuned and visual response projection maps. *Top*: the *red* and *blue colours* encode leftward and rightward gaze-tuning $Z$-score, respectively ($N = 7$ fish). *Bottom*: the *red* and *blue colours* encode the visual response $Z$-score (Supplementary Methods) to unilateral stimulation on the left and right eye, respectively ($N = 11$ fish). The HBO circuit (*white rectangle*) is engaged in both sensory and motor processing. **c** Coronal sections of the visual response map along the *two dotted lines* shown in (**a**). In (**b**) the *dotted rectangle* delineates the recorded volume. **d** Alternated unilateral visual stimulation. **e** Example traces of left and right HBO. **f** Trial-averaged response of the HBO over 20 stimulation periods. *Shaded regions* correspond to left (*dark grey*) and right (*pale grey*) illumination. **g** Bilaterally symmetric 100 ms-long flashes. **h** Example traces of the right and left HBO. The *grey lines* indicate the flashes. **i** Trial-averaged flash-induced responses of each subpopulation (100 flashes). **j** Left vs. right HBO responses (1311 flashes, $N = 12$ fish, $r = -0.5$). **k** Phase-dependent response of each subpopulation to symmetric flashes. **l** Time-evolution of the HBO oscillatory phase $\varphi(t)$. The slope of the *green line* corresponds to the stimulation frequency (period $T_{stim} = 10$ s). The HBO is entrained at half the frequency of the stimulation (period $2T_{stim}$). **m** PDF of the HBO stimulation phase offset $\delta\varphi$. The *inset* shows the PDF of the HBO phase at times where the flashes were delivered. Notice that in (**f**), (**i**) and (**k**), the *error bars* indicate the s.e.m

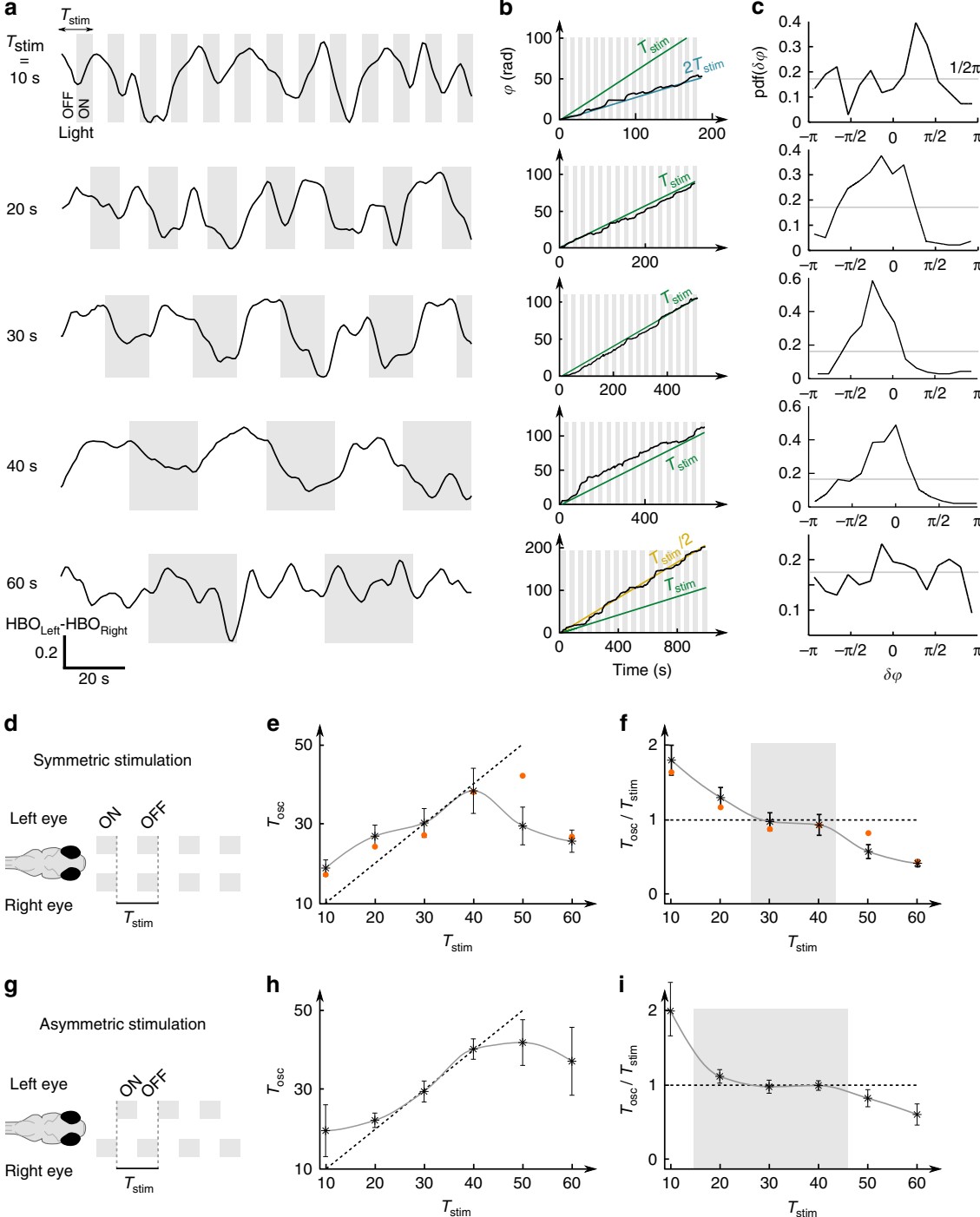

**Fig. 5** Phase-locking and frequency entrainment of the HBO by periodic visual stimuli. **a** Example HBO signals during symmetric visual stimulus presentation consisting of alternating light-ON and light-OFF periods of similar duration. Difference of the left and right HBO signals for five stimulation frequencies. Light-ON periods are indicated by the *shaded areas*. **b** Time-evolution of the HBO phase signal demonstrating frequency entrainment by periodic visual stimuli. The *green line* indicates the phase evolution of the stimulation. The HBO is entrained at half the frequency for $T_{stim} = 10$ s, and twice the frequency for $T_{stim} = 60$ s. **c** The HBO stimulation phase offset displays a non-uniform distribution, revealing phase-locking (Rayleigh test, $p < 0.01$). **d–f** Frequency entrainment by symmetric stimulation. **d** Symmetric stimulation schematic. **e** HBO oscillatory period $T_{osc}$ as a function of the stimulation period $T_{stim}$ ($N = 5$ fish, *red points* show the data presented in (**a–c**)). The *dotted line* shows the $T_{osc} = T_{stim}$ entrainment curve. **f** Entrainment ratio $T_{osc}/T_{stim}$ as a function of $T_{stim}$ ($N = 5$ fish). **g–i** Frequency entrainment by asymmetric stimulation ($N = 11$ fish). The *shaded areas* in (**f**) and (**i**) are guides to the eye showing the approximate domain of one-to-one frequency entrainment in each configuration. In (**e**, **f**) and (**h**, **i**), the *error bars* represent the s.e.m

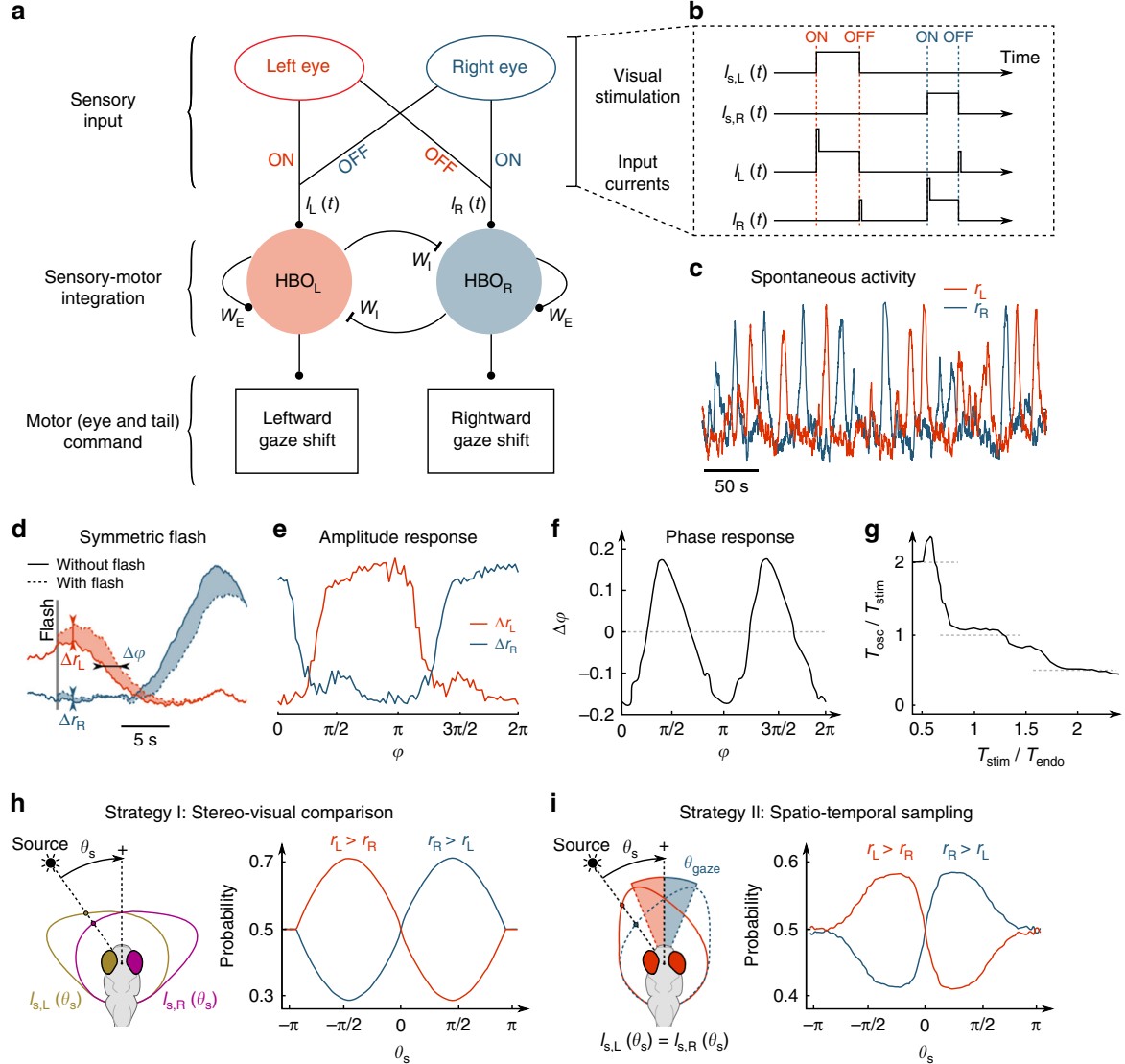

**Fig. 6** A rate model of the HBO's visually entrained dynamics leading to phototaxis. **a, b** Schematic of HBO network connectivity. Visual stimuli are relayed to the HBO via the ON and OFF pathways, as detailed in (**b**). The HBO activity controls gaze shifts via ocular saccades and turning bouts. Self-oscillatory dynamics result from the recurrent excitation ($W_E$), reciprocal inhibition ($W_I$) and adaptation currents (Supplementary Methods). **c** Example traces of the endogenous antiphasic oscillations of the left and right HBO in the absence of varying visual stimulation (arbitrary units). **d** Response of the HBO to a symmetric flash. The *shaded areas* display the difference in activity induced by the flash, from which we extract an amplitude response for each circuit and a flash-induced phase offset. **e, f** Mean amplitude and phase response to symmetric flashes as a function of the HBO phase at stimulation. **g** Frequency entrainment curve for symmetric periodic stimulations (light-ON–light-OFF alternation). The ratio of the oscillatory over the stimulation periods ($T_{osc}/T_{stim}$) is plotted as a function of $T_{stim}$ normalised by the endogenous period $T_{endo}$. **h** Numerical implementation of a stereo-visual comparison strategy for phototaxis. Each eye receives a different intensity ($I_L(\theta_s)$ and $I_R(\theta_s)$) set by the light source azimuth $\theta_s$. The *red* and *blue curves* show, as a function of $\theta_s$, the time fraction for which $r_L > r_R$ and $r_R > r_L$, respectively. The light source reinforces the ipsilateral circuit leading to orientational positive phototaxis. **i** Spatio-temporal sampling strategy. Both eyes receive a similar intensity, which depends on the relative orientation of the gaze $\theta_{gaze}$ and the light source azimuth $\theta_s$. The gaze orientation oscillates between two values according to $\theta_{gaze} = \text{sign}(r_L - r_R) * 15°$. As in (**h**), the left vs. right bias of the HBO as a function of $\theta_s$ is consistent with positive phototaxis

To evaluate the HBO's putative role in driving the phototactic behaviour, we sought to investigate whether its endogenous dynamics could be visually biased. We thus successively probed its response to asymmetric (unilateral illumination of one eye) and symmetric (similar illumination on both eyes) visual stimuli (Fig. 4). To prevent visual perturbations by the light-sheet, these experiments were carried out using two-photon excitation, i.e. with an infrared source that is invisible to the fish[28]. In a first series of experiments, we alternately illuminated each eye for 15 s while performing brain-wide functional imaging. The associated response maps indicated that the most prominent visually sensitive clusters in the hindbrain corresponded to the HBO

(Fig. 4b–f; Supplementary Fig. 5b). The right HBO, whose activity is associated with a rightward reorienting bias, was activated when the right eye was stimulated, and vice-versa. These unilateral visual responses are consistent with a stereo-visual positive phototaxis mechanism. In 1 of 11 fish, however, we found a reverse response, i.e. the visual stimulation elicited activity in the contralateral HBO. This observation parallels, and may explain, the 10% of fish found to display negative phototaxis in the behavioural assays.

Phototaxis has been shown to implicate OFF retinal ganglion cells, i.e. neurons activated upon light decrement[12]. We thus asked whether the visually evoked transient of the left or right

HBO was driven by the stimulus onset (light-on) on the ipsilateral eye, the stimulus offset (light-off) on the contralateral eye, or both. To answer this question, we monitored the HBO dynamics while exposing one eye to alternating on–off illumination, the other being maintained at constant illumination. We found that light-on and light-off stimuli induced a sharp increase in activity in the ipsilateral and contralateral sub-populations, respectively. This indicates that the ON and OFF pathways both contribute to driving the HBO (Supplementary Fig. 7, $N = 8$ fish).

In a second experiment, we investigated the HBO response to bilateral illumination by delivering a series of 100 ms-long flashes simultaneously to both eyes with a 10 s inter-flash period (Fig. 4g, h). Both left and right HBO displayed a significant response as evidenced by the trial-averaged peri-stimulus signal (Fig. 4i, $n = 1311$ flashes, $N = 12$ fish, Supplementary Fig. 7d, e), but the across-trials dispersion appeared surprisingly large (coefficient of variation 170%). The visual responses of the left and right HBO were strongly anti-correlated ($r = -0.5$) such that a given flash seemed to stochastically evoke a response in either one of them (Fig. 4j). To evaluate whether this apparent stochasticity may reflect a state-dependent sensitivity of the HBO, we examined how the stimulus-evoked transient depended on the particular phase of the oscillation at which the stimulus was delivered. We used the Hilbert transform to extract the phase $\varphi(t)$ of the HBO oscillation. We then computed the derivative of the left and right HBO signals at all times corresponding to a given phase, separating those at which a stimulus had been delivered from the times when the oscillation was running free. We then subtracted the means of both groups to extract the sole contribution of the stimulus to the HBO transient dynamics. This analysis yielded a phase-dependent response curve for each left and right HBO (Fig. 4k). It revealed that each subpopulation is responsive to the visual stimulus within a particular phase range and is essentially unresponsive for the rest of the cycle. More specifically, each flash selectively reinforces the cluster that is already active or about to become active after a period of rest. The transition between left–right HBO activity being associated with gaze shift events, such a phase-dependent response curve can be interpreted as a motor-related gating mechanism. In this view, each subpopulation selectively integrates light increments generated by ipsiversive saccades as expected during spatio-temporal sampling.

**HBO entrainment by periodic uniform visual stimuli**. In biological and physical oscillators, phase-dependent responses can be revealed through phase-locking and frequency entrainment by periodic forcing[29–31]. Consistently, we observed that the HBO oscillation exhibited a mean period of 20 s in the symmetric stimulation experiments, i.e. precisely twice the stimulation period $T_{stim} = 10$ s (Fig. 4l, $T_{osc} = 20.5 \pm 1.4$ s, $N = 12$ fish). Furthermore, the phase offset between the HBO and the stimulus signal, defined as $\delta\varphi(t) = \varphi(t) - \pi * t/T_{stim}$, displayed a marked bimodality, with two peaks located at 0 and $\pi$ (Fig. 4m, Rayleigh test, $p < 0.01$, Supplementary Fig. 8a). This indicates that visual stimuli induce a phase-shift in the HBO oscillations such that the flashes tend to coincide with successive right-to-left and left-to-right transitions (Fig. 4m inset).

We suspected that this period-doubling mechanism ensued from the particular choice of the stimulation period, which is close to half the HBO endogenous period $T_{endo}$. We sought to explore how this synchronization process varied with the stimulation $T_{stim}$. To guarantee that the mean illumination was independent of the stimulation period, we exposed the fish to a series of alternating ON–OFF illuminations at six frequencies

(20 periods each, $10\,s < T_{stim} < 60\,s$) separated by 100 s-long periods of constant illumination (Fig. 5a). For each sequence, we extracted the mean oscillation period $T_{osc}$ and the phase-offset distribution (Fig. 5b, c). In all tested visually responsive fish ($N = 5$ of 7), this analysis revealed the existence of an entrainment plateau at intermediate stimulation periods ($20\,s < T_{stim} < 40\,s$) for which the HBO frequency was controlled by the stimulation period ($T_{osc}/T_{stim} \cong 1$) and the phase-offset distribution was strongly non-uniform (Fig. 5d–f, Rayleigh test, $p < 0.01$). At higher and lower frequencies, the entrainment ratio $T_{osc}/T_{stim}$ was close to 2 and 0.5, respectively. The same experiments were replicated for asymmetric stimuli and yielded qualitatively similar results, albeit with a more pronounced phase-locking and a more extended entrainment plateau (Fig. 5g–i, $N = 11$, Supplementary Fig. 8).

**A stochastic neural model of phototaxis**. A rate model approach[32] was used to model the oscillatory dynamics of the HBO and its visually evoked responses based on the connectivity architecture suggested by our experimental observations (Fig. 6a). We described the HBO as a half-centred oscillator consisting of two symmetric modules with recurrent excitation, reciprocal inhibition, and adaptation currents (Supplementary Methods). Such a simple model, derived from a spiking, conductance-based model, has been proposed for cortical circuits to account for the phenomenology of perceptual bi-stability[33, 34]. This description is consistent with the neurotransmitter identity of the different clusters that participate in the self-oscillatory dynamics[25]. The medial clusters of the HBO were identified as glutamatergic and the lateral clusters as primarily GABAergic, reflecting the hindbrain columnar organisation of the neurotransmitter classes[35].

The model parameters were adjusted to best reproduce the self-sustained quasi-periodic oscillations observed in the absence of visual input (Fig. 6c; Supplementary Fig. 9). Visual inputs were assumed to be relayed to the HBO in the simplest way consistent with the proposed connectivity (Fig. 6b). A step increase in the illumination of the right eye was associated with a burst of current in the right circuit, followed by a plateau, whereas a step decrease in the illumination induced a transient current in the left circuit to account for the OFF pathway contribution. We then numerically implemented the stimulation protocols used in the functional imaging experiments. The left and right circuit responded to ipsilateral asymmetric visual stimuli, but we further observed a phase-dependent intensity response curve to symmetric flashes in qualitative agreement with its experimental counterpart (Figs. 4k and 6d, e). The model allowed us to reinterpret the latter result in terms of a phase-response curve (PRC). Within this scheme, the effect of a stimulus is to evoke a phase offset, i.e. a transient slowing down or speeding up of the oscillatory dynamics. For each phase, we thus computed the visually induced phase delay $\Delta\varphi$ by comparing the phase evolution after stimulation and in the free-running (without stimulation) regime (Fig. 6f). The resulting $\pi$-periodic PRC provides a direct interpretation of the phase-locking mechanism around the phases $\pi/2$ and $3\pi/2$, where the PRC displays a negative slope[30]. Consistently, a phase-locking mechanism was observed with an entrainment plateau around the endogenous oscillation frequency (Fig. 6g). However, for this process to quantitatively compare with the experimental data, the current noise had to be set at a relatively low value (Supplementary Fig. 10).

We sought to evaluate whether this minimal neural model could account for the orientational phototactic behaviour under the assumption that an imbalance in the HBO left vs. right

activity distribution may be viewed as a proxy for the statistical orientational bias in the animal locomotion[25]. We considered configurations in which a distant light source was oriented at an angle $\theta_s$ with respect to the animal's rostro-caudal axis. The total illumination impinging on each eye was taken as a slowly varying function of $\theta_s$, reflecting the large visual field of view of the zebrafish, with a maximum at $\theta_s = 80°$ for the right eye (Fig. 6h; Supplementary Methods). We successively considered the stereo-visual comparison and spatio-temporal sampling phototactic strategies. In the first case, we examined the sole effect of a continuous unbalanced bilateral illumination. For each angle $\theta_s$, the illumination on each eye was converted into a constant input current on the ipsilateral subcircuit. We then numerically computed the HBO dynamics and compared the mean left vs. right activity. We found the HBO oscillations to be statistically biased towards the subnetwork ipsilateral to the light source. This was assessed by comparing both the time fraction during which one circuit was more active than the other (Fig. 6h) and the mean activities of the two circuits (Supplementary Fig. 11). Within the assumption that such an imbalance reflects a reorienting bias[25], this result appears consistent with a positive phototactic mechanism.

In a second approach, we focused on the spatio-temporal sampling strategy (Fig. 6i). We imposed identical illumination on both eyes at all times so as to provide no explicit spatial cue to the circuit. The stimulus intensity was assumed to be a slowly decreasing function of the gaze orientation relative to the light source. We posited that each left-to-right or right-to-left transition of the HBO corresponded to a rightward or leftward gaze shift of amplitude 30°, respectively. This ±15° gaze oscillation in turn elicited a series of abrupt changes in the perceived stimulus. This procedure yielded a left vs. right activity bias that depended on the light source azimuth (Fig. 6i; Supplementary Fig. 11) and was again consistent with a positive phototactic behaviour.

## Discussion

A central goal in neuroscience is to decipher the neural computation underlying the execution of natural tasks. In this respect, whole-brain functional imaging constitutes a game-changing technique because it allows one to systematically map entire neuronal populations that respond to a given sensory stimulus[7, 28, 36] or correlate with a particular motor pattern[20, 25]. Here, this strategy was taken one step further: we used three distinct functional properties as joint criteria to identify a sensorimotor hub likely to play a key role for phototaxis. On the basis of behavioural assays, we reasoned that there should exist a circuit that both sets the pace of spontaneous saccades and integrates visual stimuli. We thus successively delineated three neuronal populations that (i) correlated with the gaze signal during spontaneous saccades, (ii) displayed self-sustained oscillating activity in eye-fixed configurations, and (iii) responded to unilateral visual stimulation. We then used a morphological registration algorithm to demonstrate the existence of a well-defined intersecting neuronal ensemble in the rostral hindbrain, identified as the HBO, that combines all three functionalities (Supplementary Movie 4).

Using a rate model, we showed that the HBO's phase-dependent visual sensitivity suffices to explain the tendency of larvae to reorient towards illuminated areas using both stereo-visual comparison and spatio-temporal sampling. The proposed sensorimotor processing leading to phototaxis can be summarised as follows: (i) the HBO self-sustained oscillations drive a quasi-periodic sequence of gaze shifts that allow the animal to actively explore the light angular gradient; (ii) asymmetric visual stimuli enhance the activity of the HBO clusters ipsilateral to the eye that receives the larger illumination, which in turn biases the fish reorientation towards the light source; and (iii) a gaze shift-induced light increment reinforces the active HBO subpopulation (ipsilateral to the new gaze direction), thus delaying further left–right transition. Over multiple cycles, this process yields a statistical bias of the fish turning probability towards the brightest region.

In foveate species, ocular saccades enable detailed visual exploration by sequentially bringing different regions of the visual scene onto a small area of the retina (the fovea) where the resolving power is maximal[37]. The present study suggests that saccades in afoveate species such as zebrafish may subserve a non-image forming visual function by allowing the animal to sequentially sample the light level at two distinct angles without the need to execute a whole-body reorientation. Within this scheme, the large stereotyped gaze shifts that characterize spontaneous saccades in zebrafish might be understood as a way to maximise the detected contrast in the context of weak illumination gradients.

The HBO presents striking similarities with central pattern generators (CPG). These specialized networks drive cyclic motor patterns underlying rhythmic behaviours such as locomotion, chewing, or breathing[38]. Although they can operate in the absence of external sensory or proprioceptive inputs, the latter have an important role in regulating the circuit dynamics to produce behaviourally relevant motor output[39]. Their action generally varies with the particular phase within the cycle at which they occur[40]. The HBO differs from standard CPGs in the sense that it drives a pseudo-periodic motor pattern—the saccadic period displays significant variations from cycle to cycle—but also controls the sequence of discrete turning-bout orientations. By modulating the rhythmic HBO oscillations, the visual inputs thus modulate the temporal pattern of reorienting movements, effectively biasing the animal's trajectory towards brightest areas over multiple cycles. It has been recently demonstrated, in *Drosophila* larvae, that sensorimotor neurons can be engaged in distinct goal-directed behaviours[41]. It will thus be interesting to see whether other sensory information—chemical cues, water flow, temperature gradients, etc.—may also be relayed to the HBO, and how these multi-sensory cues are weighted in order for the larva to navigate towards the most favourable environments.

Beyond phototaxis, the sensorimotor processing operated by the HBO may also be examined in the broader context of active sensing. In vertebrates, including mammals, sensory perception—olfaction, touch, vision, etc.—generally rely on conjugated motor routines[42, 43] such as sniffing, whisking, eye saccades, etc. The rhythms of these motor patterns have been shown to entrain low-frequency neuronal oscillations in primary sensory cortices, which in turn modulate their sensitivity according to specific motor phases[44]. This sensorimotor coordination has a key role in sensory processing. Given the generic nature of this neural mechanism, it will be exciting to further investigate whether the HBO oscillation may similarly drive, through afferent projections, rhythmic changes in sensitivity in sensory centres such that the olfactory bulb or the optic tectum.

**Data availability**. All data and codes used for the analysis are available from the authors on request.

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

## Acknowledgements

We are very grateful to Claire Wyart for allowing us to use her optogenetic setup. We would like to thank the IBPS fish facility staff, and in particular Stéphane Tronche and Alex Bois for their invaluable help. We are also grateful to Carounagarane Dore for his important contribution to the design of the light-sheet microscope. This work was supported by C'Nano Ile de France, École des Neurosciences de Paris Ile-de-France (ENP), Fondation pour la Recherche Médicale (FRM: FDT20160435670 and FDT20150532639) and Fondation Pierre-Gilles de Gennes (FPGG).

## Author contributions

S.W., A.D., V.B., R.C., R.M. and G.D. designed the project. S.W., T.B., V.B., R.C., S.K. and G.D. carried out the behavioural and neuroimaging experiments and analysis. D.G.C.H. and I.H.B. generated the elavl3:GCaMP6f transgenic zebrafish line. A.D. and R.M. developed the model and performed data analysis. S.W. and U.L.B. performed the optogenetic assays. All the authors contributed to writing the manuscript.

## Additional information

**Competing interests:** The authors declare no competing financial interests.

