## [Peer Review File · Nature Communications]

Reviewers' comments:

Reviewer #1 (Remarks to the Author):

Review of Wolf et al.; A sensorimotor hub driving phototaxis in zebrafish

The authors identify a neural network in the zebrafish brainstem that is involved in saccadic eye movements and orienting movements in a manner consistent with phototaxis. They first show that animals exposed to different types of visual input make orienting responses that are consistent with phototactic behavior, either by sampling a uniform visual field or by reorienting to a more intense visual stimulus. They then examine the correlations between orientation shifts and the activity in different regions of the brain stem, identifying populations of neurons that are strongly related to saccades. A subset of these areas also show spontaneous oscillatory activity and this activity can be entrained by either visual stimulation or optogenetic stimulation. They propose that this network in the rostral network is a Saccade Pattern Generator (SPG) that produces coordinated gaze shifts for phototactic behaviors. In general, this appears to be a very nice set of experiments and the results are generally well described through the paper.

Major comments

The analyses in the paper seem to mainly take a region of interest approach to analyzing the data, akin to a MRI study. This is generally fine, but it leaves a somewhat incomplete view of how these regions might work. For instance, comparing the gaze tuned vs. oscillatory networks in Figure 3a (assuming that I'm interpreting things correctly), it appears that the distribution of activated neurons, although overlapping, are not identical in the two conditions. This can also be seen comparing 3b to Figure 2. The same issues are present when examining the activation of neurons to visual input shown in Figure 4b and comparing them to the gaze tuned neurons. These comparisons raise the issue of whether the same neurons are involved in each of these conditions – saccade related activity (potentially also tail movements, though that's probably too much movement artefact), oscillatory activity, and visual responses. This would ideally be resolved by examining the same neurons in each of these conditions, but even evaluating partial overlaps more directly would be helpful. It would seem that the authors have this data and including these analyses might provide a finer grained description of these networks.

From the figures, it appears that the authors are identifying the SPG with the gaze related areas in the rostral hindlimb located in rhombomeres 1,2 and 3. Yet the areas don't appear to function the same across the different manipulations. For example, the box in Figure 3a indicates the SPG regions, including rhombomeres 1-3. But in Figure 3i where they examine whether optogenetic stimulation entrains eye movements, it is striking that stimulation of the area in rhombomere 1 doesn't entrain saccades whereas stimulation of the other regions do. I might have missed it, but I didn't see this discussed in the text but would seem to be an important distinction between these regions. Is such a difference predicted from the modelling?

Line 187 says that optogenetic stimulation of the SGBN consistently entrained saccadic dynamics, but this was only observed in 8/19 animals. This doesn't seem so consistent to me. Why the variability across animals? Also, the language here seems to imply that they only looked at effects of rh2-3 stimulation in the 8 animals that were entrained by stimulation in the SGBN. Were the 11 other animals tested and were also negative?

In the discussion there are several references to the study showing 'optimal' phototaxis. This seems a stretch from the current experiments, though examining this more directly would be very interesting. For instance, in an optimal sense one might expect larger or more frequent gaze shifts if the light

gradient were diffuse whereas a more punctate light source might produce smaller and less frequent gaze shifts. I.e. the uncertainty of the light source location should modulate the searching behavior of the animals. Showing such an effect in the behavior, having it be reflected in the identified networks, and captured by the simulated networks would be a very nice addition to the paper, though simply being more careful with the 'optimal' language would probably also suffice.

Minor comments

Line 112 indicates that 10% of fish show negative phototaxis, which is very interesting, though puzzling. With regard to the results of this study, this might suggest that a similar percentage of animals should show unexpected activation of brain stem neurons to light stimulation. Was this ever observed?

Line 152, what does 'most salient' mean here?

Line 171, The lower activity in the SGBN when the eyes are fixed and the animal paralyzed doesn't necessarily mean there's no fictive eye movement. The absence of afferent feedback might reduce activation of these neurons for a variety of reasons. Maybe just drop this point? It doesn't seem fully necessary.

In the paragraph starting on line 192, where they talk about the apparent long lead burst neurons in the caudal hindbrain, they suggest that these might be inhibited by the contralateral SPG. As far as I understand it, this is simply based on the observation of the average time courses of their activity. Given that so many signals are covarying, this conclusion might be strengthened by examining the trial by trial correlations, evaluating whether there is earlier build up when activity in the iSPG is lower earlier, etc... If this build up activity were due to a released inhibition from the ipsilateral SPG, there might be a consistent relationship between the activation levels in the iSPG and these regions.

Line 217 '...turning bouts orientation...' seems off. Should it be '... why the orientation of a turning bout..'?

Line 381-5, I'm not sure I understand these points. How do these results suggest an 'encephalization of CPGs'? The SPG network examined here is located near to the motor neurons that it controls, in the same way that CPGs in the spinal cord are. I also don't understand how these results illustrate stochastic vs. deterministic control.

Figure 5 title: 'frequency'

Line 506 'entrainment'

Line 682 One of the subscripts on theta here is presumably wrong?

Line 688 Does binarized mean digitized?

Equation near line 707 Why was the 30s window used to integrate the gaze shifts?

Line 781 The velocity signal used here is fine. However, it clearly misses any differences in velocity amplitude across saccades, which might provide further information about correlation. Why wasn't velocity taken directly from differentiating the position signal?

Line 892 Here activity corresponds to a firing rate, indicated by rL/R? This could be stated more clearly.

Line 929 'Modelled'

Reviewer #2 (Remarks to the Author):

This interesting study investigates saccade dynamics and its relationship with the neural circuitry driving phototaxis. First, the kinematic features of spontaneous saccades and associated tail beats are characterized. Then, saccades (gaze shifts to brighter regions in the visual field) are studied in the context of phototaxis. Light-sheet imaging is used to image a large volume of the zebrafish brain. A generalized linear model is applied to extract neuronal population tuning to saccade events (speed and position). This highlights four brain regions, one of which is the previously described hindbrain oscillator circuit. Functional mapping via single photon structured photostimulation is used to identify brain regions whose activation is associated with saccades. Functional imaging (2P) in 10 planes is performed to characterize the brain dynamics during periodic visual stimulation. Last, it is shown that the self-oscillatory dynamics can be entrained by the frequency of stimulation. A second order simulation of the circuit dynamics is developed based on a parsimonious model of the circuit. The data are interesting and important. However, the manuscript should be extensively revised to distill the safe conclusions and separate them from the more speculative interpretations.

General remarks:

- The story is hard to follow, as it jumps from hypothesis to hypothesis and sometimes switches experimental paradigms. The authors should invest some effort into improving the writing.
- The figures are not properly balanced. Some of the figures should be simplified by removing non-essential plots. The plots should be replaced by quantifications in the figure legend or in the main text. Some of the contents of the extended data should be moved to the main figures.
- Figure legends are insufficiently detailed. In some cases, I had to guess what some of the panels were supposed to show.
- Nomenclature is often inconsistent. This should be fixed to improve readability.
- Parts of the Results are speculative. All interpretations not directly supported by data should be removed or moved to the Discussion.

Specific major points:

1) Claims made in the title are not justified. The relation between visually driven gaze reorientation and tail movements need to be analyzed more closely. The distribution of the saccades shown is not bimodal as claimed. One can easily spot lower amplitude re-orientation events close to $g=0$. In fact, these low-amplitude saccades are the ones that are best correlated with tail dynamics (see Fig. 1E, F). The authors need to characterize these two components of the saccade population: the bimodal component not really correlated with tail movements and the less frequent, but better correlated, low-amplitude component. Figure 1D pools all the events, losing important details: a component with long lasting fixation events (corresponding to large amplitude saccades, self oscillating) and a component with shorter fixation time (and lower g amplitude, maybe visually driven). The authors need to show a scatter plot with g vs. fixation time. It is not clear how the correlation in (m, g) space is evaluated. What are the inclusion/exclusion criteria? Data shown in Ext. Data 1D, E should be moved to the main figure.

2) The phototaxis paradigm with two LEDs is poorly explained. Are all the saccades included? What is the proportion of saccade events that are associated with single sided illumination, and what is their g ?

3) Important information is missing from the functional imaging experiments: three main figures are

showing the very same averaged activity map associated with spontaneous saccades, but all of them miss the critical point of showing the brain region analyzed by the GLM. Regions like tectum, pretectum, cerebellum, and nMLF seem not to have been included. The single fish functional maps should be shown in the Extended Data, not just the averaged brain map. Are authors comparing one-photon lightsheet data at 488nm with 2P lightsheet data at 930nm? If so, how can they exclude a spurious activity component/bias due to the change in the background light level between the two conditions? Is the brain used as reference acquired with lightsheet or coming from a different stack acquired with other techniques?

4) The authors claim that the same circuit supports spontaneous exploratory saccades and gaze-reorienting saccades. To prove this point, they need to perform imaging experiments and run the GLM with refined regressors that distinguish between spontaneous saccades and visually induced gaze shifts.

5) "The rostral hindbrain gaze-tuned network being the sole circuit exhibiting such a property." This sentence is not supported by experimental data, because the authors didn't image everywhere. Also, in their ChR2 experiments the authors do find a region in the midbrain with very high saccade score. This region is ignored in the text.

6) For the ChR2 experiment, the photostimulation volume needs to be evaluated.

7) There is no clear evidence provided that the previously discovered SPG drives phototaxis as the title claims. Evidence is provided that the SPG is one of the nuclei whose activation drives saccades and that the SPG is active during spontaneous saccades.

8) Interestingly, the SPG circuit can be entrained for a certain frequency range. However, the visual stimulation protocol used is different from the one used for phototaxis (which does not lead to comparable entrainment levels). Furthermore, it is not clear if this one-side stimulation paradigm or the one called alternate are actually engaging a phototaxis motor program including fish tail reorientation. This should be evaluated using behavioral experiments.

Reviewer #3 (Remarks to the Author):

This paper shines a new light, so to speak, on the potential role of a striking and important set of oscillating neurons in the hindbrain of zebrafish, first reported by the Ahrens lab. There is much to like about the paper, which reveals a correlation between these neurons and saccade generation (their causation is weaker) and makes a case for their involvement in visual based spatial navigation. I have some concerns.

1. I may have missed it, but it is important to know whether the population of neurons is active in EVERY saccade. It is in the examples shown, but some indication of whether it seems to be involved in every saccade (or nearly so, given the vagaries of imaging) makes a difference. If it is not always active when there is a saccade, then their interpretation is more complicated.

2. The optogenetic activation of a region in a transgenic line with all neurons labeled obviously does not pin the saccade generation on the specific cells in which you saw activity in other experiments. It says it could potentially be them, but it could be many others, including ones with processes in the region whose somata are elsewhere. Minimally, a forthright account of this issue is warranted. Ideally, some more specific activation is needed. The "driving" used in the title depends on this non-ideal experiment.

3. I don't think that a name change of the region is warranted. I prefer the previous more generic, less function laden name hindbrain oscillator simply because even in this case we do not know for sure

what exactly this region does, as we have no circuitry (in spite of using the term network in the paper), but just some active cells correlated with some features of behavior. Maybe down the road with more connectivity information, it will be warranted.

4. The use of the word network is a bit loose. To me, a network implies information about connectivity, but there is no connectivity information, rather correlated activity in groups of neurons. I know their approach is the way it is used in fMRI, but one can actually reveal connections in non humans, so I would prefer more caution in the use of network (and circuit).

5. Extended data figure 3 refers to neurons in the title and has individual neuron numbers in the figure. Are these actually neurons and how do they know? Most of the paper is voxel based, but individual neurons appear here. Some clarity about how these individual neurons were defined is needed.

6. The paper raises obvious questions about the neuronal classes involved. Given the columns and the known columnar organization of cell types in hindbrain defined by the Fetcho lab and the mapping of other eye related neurons onto them by the Aksay lab, it seems some mention of that work which provides an obvious path to revealing a circuit is warranted. In fact, they may be able to simply cross into available transgenic lines and tell what neuronal types are involved, because their columns are oriented like the ones previously described. In addition, their idea about hindbrain oscillators arising from spinal ones makes sense. Given that the columns identified in prior work extend all the way into hindbrain, it should be possible to ask if the same cell classes are involved in both.

Minor:

Line 60: in a number of taxa

Line 61: Higher vertebrates, though commonly used, is not really a proper term as none are higher than others. All existing vertebrates are at the tips of branches of a tree. Maybe just say but also among humans.

Line 279: controlled

Figure five title: Frequency not frequent

Line 335: a series of

Line 337 and was

Line 346: I don't think you single out a network. You found some correlated cells (who knows how many others might be in the actual network). Your work is still interesting and important, but some calibration of what it means to reveal a network or circuit is warranted by you (and others doing zebrafish work!).

Line 387: taxes behavior??

Line 407: kinetic

Line 408: eye

Line 688: bout

Line 991: sample

Line 929: modeled

Line 969: a series

Thanks for the line numbers, which make reviewing easier.

Reviewer #1 (Remarks to the Author):

Review of Wolf et al.; A sensorimotor hub driving phototaxis in zebrafish

The authors identify a neural network in the zebrafish brainstem that is involved in saccadic eye movements and orienting movements in a manner consistent with phototaxis. They first show that animals exposed to different types of visual input make orienting responses that are consistent with phototactic behavior, either by sampling a uniform visual field or by reorienting to a more intense visual stimulus. They then examine the correlations between orientation shifts and the activity in different regions of the brain stem, identifying populations of neurons that are strongly related to saccades. A subset of these areas also show spontaneous oscillatory activity and this activity can be entrained by either visual stimulation or optogenetic stimulation. They propose that this network in the rostral network is a Saccade Pattern Generator (SPG) that produces coordinated gaze shifts for phototactic behaviors. In general, this appears to be a very nice set of experiments and the results are generally well described through the paper.

We are grateful to the referee for his/her careful reading of the manuscript. We appreciate his/her appreciation of our work and constructive comments, all of which have been taken into account in the revised manuscript (see details below).

Major comments

The analyses in the paper seem to mainly take a region of interest approach to analyzing the data, akin to a MRI study. This is generally fine, but it leaves a somewhat incomplete view of how these regions might work. For instance, comparing the gaze tuned vs. oscillatory networks in Figure 3a (assuming that I'm interpreting things correctly), it appears that the distribution of activated neurons, although overlapping, are not identical in the two conditions. This can also be seen comparing 3b to Figure 2. The same issues are present when examining the activation of neurons to visual input shown in Figure 4b and comparing them to the gaze tuned neurons. These comparisons raise the issue of whether the same neurons are involved in each of these conditions – saccade related activity (potentially also tail movements, though that's probably too much movement artefact), oscillatory activity, and visual responses. This would ideally be resolved by examining the same neurons in each of these conditions, but even evaluating partial overlaps more directly would be helpful. It would seem that the authors have this data and including these analyses might provide a finer grained description of these networks.

Following the referee's suggestions, we have re-analysed individual datasets in 2P experiments in order to more thoroughly characterize the overlap between the self-oscillatory network (in the absence of stimuli) and the visually responsive neurons. Along the same line, we ran new 1P imaging experiments with larvae whose eyes were successively tethered then free. We were thus able to characterize, on individual animals, how the self-oscillatory (eyes-fixed) and gaze-tuned populations overlap. These new experiments and analysis are reported in Extended Data Fig 3.

From the figures, it appears that the authors are identifying the SPG with the gaze related areas in the rostral hindlimb located in rhombomeres 1,2 and 3. Yet the areas don't appear to function the same across the different manipulations. For example, the box in Figure 3a indicates the SPG regions, including rhombomeres 1-3. But in Figure 3i where they examine whether optogenetic stimulation entrains eye movements, it is striking that stimulation of the area in rhombomere 1 doesn't entrain saccades whereas stimulation of the other regions do. I might have missed it, but I didn't see this discussed in the text but would seem to be an important distinction between these regions. Is such a difference predicted from the modelling?

The referee is right, and we apologize for not discussing this point in the original version. We interpret the absence of saccade upon activation of the two most rostral modules of the SPG as an indication that these two assemblies may lie downstream of the HBO's gaze-driving neurons. This point is now mentioned in the revised version of the manuscript. Since the model consider the oscillator as a single functional unit, this fine-grained difference cannot be captured by the model in its present form.

Line 187 says that optogenetic stimulation of the SGBN consistently entrained saccadic dynamics, but this was only observed in 8/19 animals. This doesn't seem so consistent to me. Why the variability across animals? Also, the language here seems to imply that they only looked at effects of rh2-3 stimulation in the 8 animals that were entrained by stimulation in the SGBN. Were the 11 other animals tested and were also negative?

The variability in the optogenetic response reflects the random variegation of the ChR2 line that was previously reported in Schoonheim et al, 2010 (ref 22). We thus used the same selection as in this article: we restricted the analysis to larvae that displayed significant ocular responses to activation of rh4-6 (z -score >1), which is known to be the region of the velocity burst neurons. Unresponsive animals according to this criterium did not show any response regardless of the targeted hindbrain region, and were thus entirely excluded from the analysis. This selection process is now more clearly stated and explained in the Methods section of the revised manuscript.

Please notice that, in response to another referee, we now provide a precise estimate of the illumination volume using Kaede-based photoconversion experiments (Extended Data Fig. 5 c-d).

In the discussion there are several references to the study showing 'optimal' phototaxis. This seems a stretch from the current experiments, though examining this more directly would be very interesting. For instance, in an optimal sense one might expect larger or more frequent gaze shifts if the light gradient were diffuse whereas a more punctate light source might produce smaller and less frequent gaze shifts. I.e. the uncertainty of the light source location should modulate the searching behavior of the animals. Showing such an effect in the behavior, having it be reflected in the identified networks, and captured by the simulated networks would be a very nice addition to the paper, though simply being more careful with the 'optimal' language would probably also suffice.

We agree with the referee that exploring whether the visual contrast modulates the exploratory motor patterns would be interesting line of study. This would require extensive dedicated behavioral assays that are beyond the scope of this manuscript.

We thus rephrased the sentence that mentioned an “optimal exploratory strategy” in the discussion section.

Minor comments

Line 112 indicates that 10% of fish show negative phototaxis, which is very interesting, though puzzling. With regard to the results of this study, this might suggest that a similar percentage of animals should show unexpected activation of brain stem neurons to light stimulation. Was this ever observed?

We indeed found 1 out of 11 fish for which the oscillatory dynamics is reversed, as is now indicated in the text.

Line 152, what does ‘most salient’ mean here?

We modified the phrasing to simply indicate the existence of position-tuned clusters in the rostral hindbrain.

Line 171, The lower activity in the SGBN when the eyes are fixed and the animal paralyzed doesn’t necessarily mean there’s no fictive eye movement. The absence of afferent feedback might reduce activation of these neurons for a variety of reasons. Maybe just drop this point? It doesn’t seem fully necessary.

We agree that this sentence is unnecessary, and we therefore removed it.

In the paragraph starting on line 192, where they talk about the apparent long lead burst neurons in the caudal hindbrain, they suggest that these might be inhibited by the contralateral SPG. As far as I understand it, this is simply based on the observation of the average time courses of their activity. Given that so many signals are covarying, this conclusion might be strengthened by examining the trial by trial correlations, evaluating whether there is earlier build up when activity in the iSPG is lower earlier, etc... If this build up activity were due to a released inhibition from the ipsilateral SPG, there might be a consistent relationship between the activation levels in the iSPG and these regions.

What we merely claim is that an inhibitory drive of the LLBN by the contralateral SPG is consistent with our data, whereas an excitatory drive by the ipsilateral SPG is not. Following the referee’s remark, we examined the relationship between the LLBN and contralateral SPG fluorescent signals on a trial by trial basis. As anticipated, we found a slightly negative correlation value (0.15), which is unfortunately not significant enough to convincingly establish this putative inhibitory projection. We think that this relatively low correlation value may be due to the long decay time of the GCaMP sensor, which tends to smooth out any fast reduction of spiking rate.

Line 217 ‘...turning bouts orientation...’ seems off. Should it be ‘... why the orientation of a turning bout..’?

We rephrased this sentence accordingly.

Line 381-5, I'm not sure I understand these points. How do these results suggest an 'encephalization of CPGs'? The SPG network examined here is located near to the motor neurons that it controls, in the same way that CPGs in the spinal cord are. I also don't understand how these results illustrate stochastic vs. deterministic control.

CPGs are generally thought to drive highly periodic motor programs engaged during locomotion for instance. Here the SPG oscillation is only weakly periodic. Furthermore, the control of the animal orientation through swim-turns is only partially under control of the SPG: the latter barely defines the orientation of the tail-beats, but not the precise moments at which they are triggered.

We agree with the referee that this aspect of the discussion is highly speculative, and we decided to remove it from the manuscript.

Figure 5 title: 'frequency'

Ok

Line 506 'entrainment'

Ok

Line 682 One of the subscripts on theta here is presumably wrong?

Yes, it is now corrected

Line 688 Does binarized mean digitized?

No, binarized means that the image is represented by a 2-value matrix whereas digitized more generally means that it is represented as a matrix with discrete values.

Equation near line 707 Why was the 30s window used to integrate the gaze shifts?

The 30s window correspond to the total duration of each asymmetric illumination period (25s of maximum gradient +5s of transition back to the uniform illumination). This value may have seemed at odds with the schematic of Figure 1j, which was thus modified to more precisely reflect the stereo-visual illumination sequence.

Line 781 The velocity signal used here is fine. However, it clearly misses any differences in velocity amplitude across saccades, which might provide further information about correlation. Why wasn't velocity taken directly from differentiating the position signal?

The derivative of the gaze signal is very noisy. We thus decided to bring down to zero the signal except at saccade times. We tested the approach proposed by the referee, by using the gaze variation amplitude instead of -1 or +1 at each saccade. The saccade amplitude distribution being relatively bimodal, this alternative approach produces essentially similar maps.

Line 892 Here activity corresponds to a firing rate, indicated by rL/R ? This could be stated more clearly.

ok

Line 929 'Modelled'

ok

Reviewer #2 (Remarks to the Author):

This interesting study investigates saccade dynamics and its relationship with the neural circuitry driving phototaxis. First, the kinematic features of spontaneous saccades and associated tail beats are characterized. Then, saccades (gaze shifts to brighter regions in the visual field) are studied in the context of phototaxis. Light-sheet imaging is used to image a large volume of the zebrafish brain. A generalized linear model is applied to extract neuronal population tuning to saccade events (speed and position). This highlights four brain regions, one of which is the previously described hindbrain oscillator circuit. Functional mapping via single photon structured photostimulation is used to identify brain regions whose activation is associated with saccades. Functional imaging (2P) in 10 planes is performed to characterize the brain dynamics during periodic visual stimulation. Last, it is shown that the self-oscillatory dynamics can be entrained by the frequency of stimulation. A second order simulation of the circuit dynamics is developed based on a parsimonious model of the circuit. The data are interesting and important. However, the manuscript should be extensively revised to distill the safe conclusions and separate them from the more speculative interpretations.

General remarks:

- The story is hard to follow, as it jumps from hypothesis to hypothesis and sometimes switches experimental paradigms. The authors should invest some effort into improving the writing.*
- The figures are not properly balanced. Some of the figures should be simplified by removing non-essential plots. The plots should be replaced by quantifications in the figure legend or in the main text. Some of the contents of the extended data should be moved to the main figures.*
- Figure legends are insufficiently detailed. In some cases, I had to guess what some of the panels were supposed to show.*
- Nomenclature is often inconsistent. This should be fixed to improve readability.*
- Parts of the Results are speculative. All interpretations not directly supported by data should be removed or moved to the Discussion.*

We thank the referee for his/her careful reading of our manuscript and his/her constructive comments. Below we address each of the different points raised in the report. We also made an effort to improve the general readability of the manuscript and the figures, the precision of the captions, and to ensure that it carries a safe message to the reader. Notice also that one of the referees required us to use the original name proposed in Ahrens *et al*, 2013 (HBO) for the SPG. We use the SPG denomination in the present document, for consistency with the report, but this circuit now appears under the name HBO in the revised manuscript.

Specific major points:

- 1) *Claims made in the title are not justified.*

We have changed the title to account for this criticism. We also have modified the text so that it does not overstate the role of this circuit for phototaxis.

The relation between visually driven gaze reorientation and tail movements need to be analyzed more closely.

In the original version, the coordination between gaze reorientation and tail movements was examined “in the absence of visual cues”: Figures 1a-g are obtained from experiments performed under whole-field constant illumination. To account for this remark and also address the referee’s last point, we have run new behavioral experiments which allows us to examine the tail/eyes coordination during phototaxis (see our response to point 8).

The distribution of the saccades shown is not bimodal as claimed. One can easily spot lower amplitude re-orientation events close to $g=0$. In fact, these low-amplitude saccades are the ones that are best correlated with tail dynamics (see Fig. 1E, F). It is not clear how the correlation in (m, g) space is evaluated. What are the inclusion/exclusion criteria? The authors need to characterize these two components of the saccade population: the bimodal component not really correlated with tail movements and the less frequent, but better correlated, low-amplitude component.

We apologize for the lack of precision in our original description of the saccadic dynamics, which lead to some confusion. This paragraph has been largely re-written to carry out the right message :

- We do not claim that the distribution of saccade amplitude is bimodal, but that the distribution of the gaze angle is bimodal, as shown in fig 1C which takes into account all the time points from the measured $g(t)$ sequence.
- Regarding the coordination mechanism: in fig 1F, each data point corresponds to one discrete tail-beat, and the associated g is merely the instantaneous gaze position in the middle of the tail-beat event. Therefore, data points near $g=0$ do not correspond to small amplitude saccades, but to tail beats occurring when the gaze is close to 0. The inclusion/exclusion criterium corresponds to the detection threshold of discrete tail-beats (see Method), and is thus unrelated to the saccadic dynamics. Ext. Data fig 1d-e now shown in Figure 1g-h, demonstrates the robust correlation that exists between gaze orientation and tail beat orientation, by separating data points into two groups : $g < 0$ (gaze to the right) and $g > 0$ (gaze to the left).
- The referee suggests to distinguish in the time-correlation analysis between what we called «reorienting saccades » versus « secondary saccades ». Reorienting saccades induce a switch in gaze orientation, from left to right or vice-versa. Secondary saccades are leftward (resp. rightward) saccades occurring when the gaze is oriented to the left (resp. right). These secondary saccades indeed tend to have a smaller amplitude, as they essentially allow to compensate for the accumulated drift. We have re-analysed the behavioral data by distinguishing between these two types of events. The results are presented in Extended Data Figure 1d-g . It demonstrates that in fact both types of saccades show similar temporal correlation with ipsiversive turning bouts.

Figure 1D pools all the events, losing important details: a component with long lasting fixation events (corresponding to large amplitude saccades, self oscillating) and a component with shorter fixation time (and lower g amplitude, maybe visually driven). The authors need to show a scatter plot with g vs. fixation time.

The Figure 1D originally displayed the distribution of fixation times (delay between any successive saccades, either reorienting or secondary). As this delay is eventually compared with the endogeneous period of the SPG, which correlates with the right/left gaze alternation, we now use only the reorienting saccades. Figure 1D thus characterizes the distribution of leftward and rightward gaze periods.

Notice that this distribution is only slightly modified when taking into account every saccade, since the secondary saccades make up only 6-7 % of the total number. Second, we found only a small negative correlation ($R=-0.08\pm 0.03$) between the amplitude of the saccade and the fixation time.

Data shown in Ext. Data 1D, E should be moved to the main figure.

We have done so, and also added several panels to the two extended data figures associated with the behavioral assays, to detail this reorienting vs secondary saccades analysis. We believe that altogether, these different analysis should now provide the reader with a clear description of the saccadic and turning bouts statistics.

2) The phototaxis paradigm with two LEDs is poorly explained. Are all the saccades included? What is the proportion of saccade events that are associated with single sided illumination, and what is their g?

To analyse the effect of single-sided illumination, we focused on the modifications of the gaze distribution that follows the extinction of the light on one eye (Fig 1j-k and Extended Data Fig2 a-h). This analysis thus does not rely on the detection of the saccades, but simply on the gaze signal (all the saccades are therefore "included"). This aspect of the analysis, as well as the experimental protocol, have been made clearer in the revised manuscript.

Following the referee's comment, we further re-analysed these data to estimate the change in saccade frequency associated with single-sided illumination. We found a 27% \pm 3.4% increase in the average fixation time towards the illuminated side, as is now indicated in the text. We also computed the distribution of the gaze-shift during whole-field and unilateral illumination (Extended Data Fig. 2h).

3) Important information is missing from the functional imaging experiments: three main figures are showing the very same averaged activity map associated with spontaneous saccades, but all of them miss the critical point of showing the brain region analyzed by the GLM. Regions like tectum, pretectum, cerebellum, and nMLF seem not to have been included.

We have now delineated in all the functional maps the regions that were analysed by the GLM. As indicated in the text, we had to restrict our analysis to the hindbrain (including the cerebellum) and the caudal midbrain when identifying the gaze-tuned circuit, as the eyes rotation induces large deformation in the most caudal regions of the brain that precludes consistent signal extraction.

The single fish functional maps should be shown in the Extended Data, not just the averaged brain map.

Single fish functional maps are now provided in Extended Data Fig 3b-e. We have re-analysed individual datasets in 2P experiments to thoroughly characterize the overlap between the self-oscillatory network (in the absence of stimuli) and the visually responsive neurons. Along the same line, we ran new 1P imaging experiments with larvae whose eyes were successively tethered then free. We were thus able to characterize, on individual animals, how the self-oscillatory (eyes-fixed) and gaze-tuned populations overlap.

Are authors comparing one-photon lightsheet data at 488nm with 2P lightsheet data at 930nm? If so, how can they exclude a spurious activity component/bias due to the change in the background light level between the two conditions?

To address this issue, we have run new experiments using 2P light-sheet while simultaneously monitoring the eyes movements (Extended Data Fig 3a). The results are consistent with what we obtained under 1P imaging, indicating that the effect of background light level conditions is not significant.

Is the brain used as reference acquired with lightsheet or coming from a different stack acquired with other techniques?

The reference brain has been obtained by registering four HuC:GCaMP6 larvae using 1P light sheet imaging. This was originally indicated in the Methods section. This information is now provided in the Results.

4) The authors claim that the same circuit supports spontaneous exploratory saccades and gaze-reorienting saccades. To prove this point, they need to perform imaging experiments and run the GLM with refined regressors that distinguish between spontaneous saccades and visually induced gaze shifts.

It is not clear to us how we may distinguish between spontaneous exploratory and gaze-reorienting saccades. The only distinction we can make is between reorienting and secondary saccades as discussed above. We did separately analyse the SPG's signal (identified through regression with the gaze signal) associated with secondary and reorienting saccades. It appears that the SPG shows a robust and comparable response to both types of events, as shown in Extended Data Figure 6d-e. We also now provide a statistical analysis on the probability of each circuit to display a significant activity increase after one saccade (Extended Data Figure 6g).

5) "The rostral hindbrain gaze-tuned network being the sole circuit exhibiting such a property." This sentence is not supported by experimental data, because the authors didn't image everywhere. Also, in their ChR2 experiments the authors do find a region in the midbrain with very high saccade score. This region is ignored in the text.

We agree with the referee that we did not demonstrate the uniqueness of the SPG as a putative saccadic command circuit, so this sentence has been removed. Regarding the

optogenetic responsive midbrain region, only two fish were found to exhibit this property. We added a sentence on this rare effect at the end of the optogenetic result section.

6) For the Chr2 experiment, the photostimulation volume needs to be evaluated.

To evaluate the photostimulation volume, we have run experiments using a Tg(HuC:gal4; UAS:Kaede) line, in which the photoconvertible protein Kaede is expressed near pan-neurally. This approach was previously shown to provide a close estimate of the illuminated volume (Arrenberg et al., 2009). The result of these new experiments is shown in Extended Data Figure 5c-d.

7) There is no clear evidence provided that the previously discovered SPG drives phototaxis as the title claims. Evidence is provided that the SPG is one of the nuclei whose activation drives saccades and that the SPG is active during spontaneous saccades.

Notice that the SPG is also responsive to light in a way that is consistent with phototaxis. Although there are converging lines of evidence pointing to this particular circuit, we agree with the referee that we cannot claim that the SPG constitutes the unique command circuit driving phototaxis. We thus have changed the title and rephrased several sentences in the text to not overstate the putative role of this region.

8) Interestingly, the SPG circuit can be entrained for a certain frequency range. However, the visual stimulation protocol used is different from the one used for phototaxis (which does not lead to comparable entrainment levels). Furthermore, it is not clear if this one-side stimulation paradigm or the one called alternate are actually engaging a phototaxis motor program including fish tail reorientation. This should be evaluated using behavioral experiments.

We agree with the referee that performing virtual reality experiments, with both the eyes and tail free under two-photon imaging will be important. This study, which will require to tackle important technical issues, is beyond the scope of this work. However, to address the referee's concern regarding coordination during phototaxis, we performed new behavioral experiments where both eyes and tail dynamics were monitored under unilateral illumination. This new set of experiments indicate that the animal does perform orientational phototaxis, as expected, and that the coordination between ocular saccades and tail movements is similarly robust even during phototaxis behavior.

Reviewer #3 (Remarks to the Author):

This paper shines a new light, so to speak, on the potential role of a striking and important set of oscillating neurons in the hindbrain of zebrafish, first reported by the Ahrens lab. There is much to like about the paper, which reveals a correlation between these neurons and saccade generation (their causation is weaker) and makes a case for their involvement in visual based spatial navigation.

We are grateful to the referee for his/her careful reading of the manuscript. We appreciate his/her appreciation of our work and constructive comments, all of which have been taken into account in the revised manuscript (see details below).

I have some concerns.

1. I may have missed it, but it is important to know whether the population of neurons is active in EVERY saccade. It is in the examples shown, but some indication of whether it seems to be involved in every saccade (or nearly so, given the vagaries of imaging) makes a difference. If it is not always active when there is a saccade, then their interpretation is more complicated.

We have re-analysed the data to show that the SPG displays a transient rise of activity after almost every saccade. We have computed, for each voxel within the SPG region, the fraction of the ipsiversive saccades for which a statistically significant activity increase is observed. We then plotted the probability distribution function of this percentage for all the HBO voxels. Interestingly, it appears that the response is as strong for re-orienting than for the secondary saccades. This new analysis is referred to in the text, and shown in Extended Fig. 6g.

2. The optogenetic activation of a region in a transgenic line with all neurons labeled obviously does not pin the saccade generation on the specific cells in which you saw activity in other experiments. It says it could potentially be them, but it could be many others, including ones with processes in the region whose somata are elsewhere. Minimally, a forthright account of this issue is warranted. Ideally, some more specific activation is needed. The “driving” used in the title depends on this non-ideal experiment.

We agree with the referee that the optogenetic assay does not constitute a definitive proof of the specific role of the SPG. It merely adds to several lines of evidence that point to this particular neuronal assembly as being essential to the phototactic process. We have modified the title and the text in order to not overstate the uniqueness of this region in the phototaxis process.

In the absence of known specific promoter for this region, we cannot be more precise in the optogenetic targeting. However, in the revised version, we have added photo-conversion experiments in order to more precisely estimate the photostimulation volume associated with the optogenetic experiments, such as the provide the reader with a way to assess the extent of the photo-activated region (see Extended Data Fig. 5c-d).

3. I don't think that a name change of the region is warranted. I prefer the previous more generic, less function laden name hindbrain oscillator simply because even in this case we

do not know for sure what exactly this region does, as we have no circuitry (in spite of using the term network in the paper), but just some active cells correlated with some features of behavior. Maybe down the road with more connectivity information, it will be warranted.

We agree with this remark. We thus decided to stick to the original nomenclature proposed by Ahrens in the 2013 article (HBO).

4. The use of the word network is a bit loose. To me, a network implies information about connectivity, but there is no connectivity information, rather correlated activity in groups of neurons. I know their approach is the way it is used in fMRI, but one can actually reveal connections in non humans, so I would prefer more caution in the use of network (and circuit).

We have modified the text accordingly.

5. Extended data figure 3 refers to neurons in the title and has individual neuron numbers in the figure. Are these actually neurons and how do they know? Most of the paper is voxel based, but individual neurons appear here. Some clarity about how these individual neurons were defined is needed.

We did use a morphological segmentation in order to identify individual neurons participating in the self-oscillatory assembly (Extended Data Fig. 4a-e) The segmentation method has been reported in a previous publication. We added a sentence in the Methods section to clarify this point.

6. The paper raises obvious questions about the neuronal classes involved. Given the columns and the known columnar organization of cell types in hindbrain defined by the Fetcho lab and the mapping of other eye related neurons onto them by the Aksay lab, it seems some mention of that work which provides an obvious path to revealing a circuit is warranted. In fact, they may be able to simply cross into available transgenic lines and tell what neuronal types are involved, because their columns are oriented like the ones previously described. In addition, their idea about hindbrain oscillators arising from spinal ones makes sense. Given that the columns identified in prior work extend all the way into hindbrain, it should be possible to ask if the same cell classes are involved in both.

The neurotransmitter identification of the different clusters that participate in the self-oscillatory dynamics have been recently obtained by Dunn et al. As anticipated by the referee, the morphology of the different HBO clusters reflects the columnar organization of the neurotransmitter populations in the hindbrain shown by the Fetcho lab. More specifically, the medial clusters of the HBO were identified as being glutamatergic, and the lateral clusters as being primarily GABAergic. This neurotransmitter diversity is consistent with the proposed half-centered oscillator description that underlies the model. We now provide this important information in the article, together with the appropriate references.

Minor:

Line 346: I don't think you single out a network. You found some correlated cells (who knows how many others might be in the actual network). Your work is still interesting and important, but some calibration of what it means to reveal a network or circuit is warranted by you (and others doing zebrafish work!).

We changed the title and the discussion to account for this remark.

Line 387: taxes behavior??

We replaced it by goal-directed behavior

Line 60: in a number of taxa

Line 61: Higher vertebrates, though commonly used, is not really a proper term as none are higher than others. All existing vertebrates are at the tips of branches of a tree. Maybe just say but also among humans.

Line 279: controlled

Figure five title: Frequency not frequenc

Line 335: a series of

Line 337 and was

Line 407: kinetic

Line 408: eye

Line 688: bout

Line 991: sample

Line 929: modeled

Line 969: a series

Thanks for the line numbers, which make reviewing easier.

We thank the referee for his careful readings. These various typos have been fixed.

Reviewers' comments:

Reviewer #1 (Remarks to the Author):

In general, the authors have done a very good job of revising the manuscript. The overall result is much more straightforward and easier to read. They have sufficiently addressed most of my concerns and I only have relatively minor issues.

My main issue is in the text in lines 176-190. I am still somewhat confused what cell populations comprise the SGBN vs. the HBO vs. the 'gazed-tuned rostral hindbrain population'. It would be very helpful here to precisely define what the authors define each of these structures as. It would also help to make clear what their new contribution is here. Are they simply elaborating previously identified structures? Are they identifying additional structures that haven't been described before? Such a statement would also help in evaluating figures throughout the manuscript.

Minor points

In their regression analysis maps (Figure 2), it appears that each pixel was either velocity or position related. Were the results that binary? Or if you made separate maps for position and velocity, would there be considerable overlap? E.g. Figure 2d-f seems to suggest that different regions considered as 'position' (2,3,4) still have quite different saccade-dynamics – i.e. neurons in 2 are more tonic whereas neurons in 3 and 4 are more phasic.

Line 254: 'We found that both light-on and light-off stimuli...'

Line 313: Maybe the more fundamental reference for half-centers with fatiguing inhibition would be Graham-Brown and spinal CPGs?

Line 765: 'refresh rate'

Line 789: '1mm diameter'

Line 790: 'The eyes region was...'

Extended Figure 3 is nice. The spacing and labelling of the subpanels could be clearer, however, so it's more obvious which images are from different fish at the same depth vs. from different depths in the same fish.

Reviewer #2 (Remarks to the Author):

The authors have addressed my concerns, and the paper should now be published.

Easy to fix: The Arrenberg et al. (2009) reference for the Kaede photoactivation (line 928) has not been added to the reference list.

Orger & Baier (Visual Neuroscience 2005) were the first to report that blue light evokes phototaxis in zebrafish. Please consider citing this study, as it is foundational to your paradigm.

Reviewer #3 (Remarks to the Author):

I am content with the changes the authors made. They made a good faith, substantive effort to address the reviewer concerns and the paper contains work that is high quality and broadly interesting.

Reviewer #1 (Remarks to the Author):

In general, the authors have done a very good job of revising the manuscript. The overall result is much more straightforward and easier to read. They have sufficiently addressed most of my concerns and I only have relatively minor issues.

We would like to thank the referee for his/her valuable input on our work. Modifications to the text (shown in red) have been made to address the referee's remaining concerns.

My main issue is in the text in lines 176-190. I am still somewhat confused what cell populations comprise the SGBN vs. the HBO vs. the 'gazed-tuned rostral hindbrain population'. It would be very helpful here to precisely define what the authors define each of these structures as. It would also help to make clear what their new contribution is here. Are they simply elaborating previously identified structures? Are they identifying additional structures that haven't been described before? Such a statement would also help in evaluating figures throughout the manuscript.

We have modified the text to clarify this point. As is now clearly stated (see page 8):

1 – the SGBN (velocity-tuned - rh4-7) and the VPNI (position-tuned rh7) have been previously described. What we report in this work is the identification of 6 other bilaterally distributed clusters in the rostral hindbrain (rh1-3) whose activity are tuned to the gaze orientation.

2 – We show that these newly identified gaze-tuned populations are part of the HBO, the self-oscillating neuronal ensemble that was originally identified by Ahrens et al using correlation analysis in eyes-fixed experiments.

3 – In all the experiments, we use this self-oscillation as a way to functionally delineate the HBO (detailed in methods and Ext. Data Fig. 4). With this definition, the HBO appears to essentially encompass the rostral hindbrain gaze-tuned clusters (rh1-3), although a few neurons in the more caudal regions (rh4-7) also participate in this self-oscillatory dynamics.

Minor points

In their regression analysis maps (Figure 2), it appears that each pixel was either velocity or position related. Were the results that binary? Or if you made separate maps for position and velocity, would there be considerable overlap? E.g. Figure 2d-f seems to suggest that different regions considered as 'position' (2,3,4) still have quite different saccade-dynamics – i.e. neurons in 2 are more tonic whereas neurons in 3 and 4 are more phasic.

The referee is correct: there is a partial overlap between the two maps, as is now mentioned in the text. We have changed Video 3 to show both maps separately in order for the reader to appreciate this fact.

Line 254: 'We found that both light-on and light-off stimuli...'

Line 313: Maybe the more fundamental reference for half-centers with fatiguing inhibition would be Graham-Brown and spinal CPGs?

Line 765: 'refresh rate'

Line 789: '1mm diameter'

Line 790: 'The eyes region was...'

Extended Figure 3 is nice. The spacing and labelling of the subpanels could be clearer, however, so it's more obvious which images are from different fish at the same depth vs. from different depths in the same fish.

We have modified the manuscript to account for these different points.